# TRANSFER LEARNING WITH PRE-TRAINED CONDITIONAL GENERATIVE MODELS

## ABSTRACT

Transfer learning is crucial in training deep neural networks on new target tasks. Current transfer learning methods always assume at least one of (i) source and target task label spaces overlap, (ii) source datasets are available, and (iii) target network architectures are consistent with source ones. However, holding these assumptions is difficult in practical settings because the target task rarely has the same labels as the source task, the source dataset access is restricted due to storage costs and privacy, and the target architecture is often specialized to each task. To transfer source knowledge without these assumptions, we propose a transfer learning method that uses deep generative models and is composed of the following two stages: *pseudo pre-training* (PP) and *pseudo semi-supervised learning* (P-SSL). PP trains a target architecture with an artificial dataset synthesized by using conditional source generative models. P-SSL applies SSL algorithms to labeled target data and unlabeled pseudo samples, which are generated by cascading the source classifier and generative models to condition them with target samples. Our experimental results indicate that our method can outperform the baselines of scratch training and knowledge distillation.

## 1 INTRODUCTION

For training deep neural networks on new tasks, *transfer learning* is essential, which leverages the knowledge of related (source) tasks to the new (target) tasks via the joint- or pre-training of source models. There are many transfer learning methods for deep models under various conditions (Pan & Yang, 2010; Wang & Deng, 2018). For instance, *domain adaptation* leverages source knowledge to the target task by minimizing the domain gaps (Ganin et al., 2016), and *fine-tuning* uses the pre-trained weights on source tasks as the initial weights of the target models (Yosinski et al., 2014). These existing powerful transfer learning methods always assume at least one of (i) source and target label spaces have overlaps, e.g., a target task composed of the same class categories as a source task, (ii) source datasets are available, and (iii) consistency of neural network architectures i.e., the architectures in the target task must be the same as that in the source task. However, these assumptions are seldom satisfied in real-world settings (Chang et al., 2019; Kenthapadi et al., 2019; Tan et al., 2019). For instance, suppose a case of developing an image classifier on a totally new task for an embedded device in an automobile company. The developers found an optimal neural architecture for the target dataset and the device by neural architecture search, but they cannot directly access the source dataset for the reason of protecting customer information. In such a situation, the existing transfer learning methods requiring the above assumptions are unavailable, and the developers cannot obtain the best model.

To promote the practical application of deep models, we argue that we should reconsider the three assumptions on which the existing transfer learning methods depend. For assumption (i), new target tasks do not necessarily have the label spaces overlapping with source ones because target labels are often designed on the basis of their requisites. In the above example, if we train models on StanfordCars (Krause et al., 2013), which is a fine-grained car dataset, there is no overlap with ImageNet (Russakovsky et al., 2015) even though ImageNet has 1000 classes. For (ii), the accessibility of source datasets is often limited due to storage costs and privacy (Liang et al., 2020; Kundu et al., 2020; Wang et al., 2021a), e.g., ImageNet consumes over 100GB and contains person faces co-occurring with objects that potentially raise privacy concerns (Yang et al., 2022). For (iii), the consistency of the source and target architectures is broken if the new architecture is specialized for

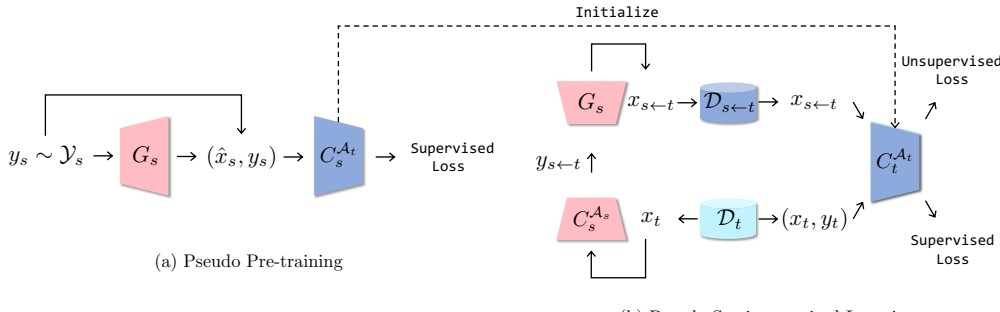

(a) Pseudo Pre-training

(b) Pseudo Semi-supervised Learning

Figure 1: Proposed transfer learning methods leveraging conditional source generative model $G_s$. Red color represents given source models, light blue represents target models and datasets, and dark blue represents the output of the proposed methods. (a) We produce initial weights of a target architecture $\mathcal{A}_t$ by training a source classifier $C_s^{\mathcal{A}_t}$ with pairs of conditional sample $\hat{x}_s \sim G_s(y_s)$ and uniformly sampled source label $y_s$. (b) We penalize a target classifier $C_t^{\mathcal{A}_t}$ with unsupervised loss derived from SSL method by applying a pseudo sample $x_{s \leftarrow t}$ while supervised training on target dataset $\mathcal{D}_t$. $x_{s \leftarrow t}$ is sampled from $G_s$ conditioned by pseudo source label $y_{s \leftarrow t} = C_s^{\mathcal{A}_s}(x_t)$.

Table 1: Comparison of transfer learning settings

|  | (i) no label overlap | (ii) no source dataset access | (iii) architecture inconsistency |
|---|---|---|---|
| Domain Adaptation | − | ✓ | − |
| Fine-tuning | ✓ | ✓ | − |
| Ours | ✓ | ✓ | ✓ |

the new tasks like the above example. Deep models are often specialized for tasks or computational resources by neural architecture search (Zoph & Le, 2017; Lee et al., 2021) in particular when deploying on edge devices; thus, their architectures can differ for each task and runtime environment. Since existing transfer learning methods require one of the three assumptions, practitioners must design target tasks and architectures to fit those assumptions by sacrificing performance. To maximize the potential performance of deep models, a new transfer learning paradigm is required.

In this paper, we shed light on an important but less studied problem setting of transfer learning, where (i) source and target task label spaces do not have overlaps, (ii) source datasets are not available, and (iii) target network architectures are not consistent with source ones (Tab. 1). To transfer source knowledge while satisfying the above three conditions, our main idea is to leverage source pre-trained discriminative and generative models; their architectures differ from that of target tasks. We focus on applying the generated samples from source class-conditional generative models for target training. Deep conditional generative models precisely replicate complex data distributions such as ImageNet (Brock et al., 2018; Karras et al., 2020; Dhariwal & Nichol, 2021), and the pre-trained models are widely used for downstream tasks (Wang et al., 2018; Zhao et al., 2020a; Patashnik et al., 2021; Ramesh et al., 2022). Furthermore, deep generative models have the potential to resolve the problem of source dataset access because they can compress information of large datasets into much smaller pre-trained weights (e.g., about 100MB in the case of a BigGAN generator), and safely generate informative samples without re-generating training samples by differential privacy training techniques (Torkzadehmahani et al., 2019; Augenstein et al., 2020; Liew et al., 2022).

By using conditional generative models, we propose a two-stage transfer learning method composed of *pseudo pre-training* (PP) and *pseudo semi-supervised learning* (P-SSL). Figure 1 illustrates an overview of our method. PP pre-trains the target architectures by using the artificial dataset generated from the source conditional generated samples and given labels. This simple pre-process provides effective initial weights without accessing source datasets and architecture consistency. To address the non-overlap of the label spaces without accessing source datasets, P-SSL trains a target model with SSL (Chapelle et al., 2006; Van Engelen & Hoos, 2020) by treating pseudo samples drawn from the conditional generative models as the unlabeled dataset. Since SSL assumes the labeled and unlabeled datasets are drawn from the same distribution, the pseudo samples should be target-related samples, whose distribution is similar enough to the target distribution. To generate target-related samples, we cascade a classifier and conditional generative model of the source domain. Specifically, we (a) obtain pseudo source soft labels from the source classifier by applying

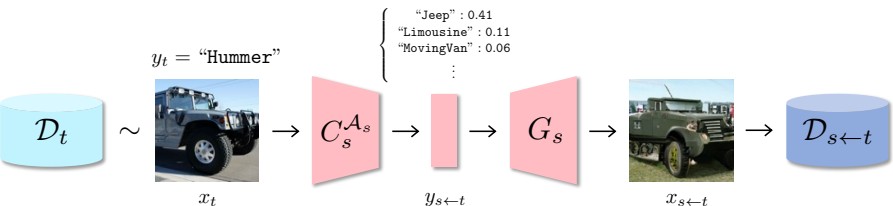

Figure 2: Pseudo conditional sampling. We obtain a pseudo soft label $y_{s \leftarrow t}$ by applying a target data $x_t$ to a source classifier $C_s^{\mathcal{A}_s}$, and then generate a target-related sample $x_{s \leftarrow t}$ from a source generative model $G_s$ conditioned by $y_{s \leftarrow t}$. In this example, $C_s^{\mathcal{A}_s}$ output $y_{s \leftarrow t}$ from the input car image $x_t$ of Hummer class by interpreting $x_t$ as a mixture of source car classes (Jeep, Limousine, MovingVan, etc.), and then $G_s$ generate a target-related car image from $y_{s \leftarrow t}$.

target data, and (b) generate conditional samples given a pseudo source soft label. By using the target-related samples, P-SSL trains target models with off-the-shelf SSL algorithms.

In the experiments, we first confirm the effectiveness of our method through a motivating example scenario where the source and target labels do not overlap, the source dataset is unavailable, and the architectures are specialized by manual neural architecture search (Sec. 4.2). Then, we show that our method can stably improve the baselines in transfer learning without three assumptions under various conditions: e.g., multiple target architectures (Sec. 4.3), and multiple target datasets (Sec. 4.4). These indicate that our method succeeds to make the architecture and task designs free from the three assumptions. Further, we confirm that our method can achieve practical performance without the three assumptions: the performance was comparable to the methods that require one of the three assumptions (Sec. 4.5 and 4.7). We also provide extensive analysis revealing the conditions for the success of our method. For instance, we found that the target performance highly depends on the similarity of generated samples to the target data (Sec. 3.2.4 and 4.4), and a general source dataset (ImageNet) is more suitable than a specific source dataset (CompCars) when the target dataset is StanfordCars (Sec. 4.6).

## 2 PROBLEM SETTING

We consider a transfer learning problem where we train a neural network model $f_\theta^{\mathcal{A}_t}$ on a labeled target dataset $\mathcal{D}_t = \{(x_t^i \in \mathcal{X}_t, y_t^i \in \mathcal{Y}_t)\}_{i=1}^{N_t}$ given a source classifier $C_s^{\mathcal{A}_s}$ and a source conditional generative model $G_s$; $C_s^{\mathcal{A}_s}$ and $G_s$ are *off-the-shelf* i.e., we do not access source datasets to pre-train them. $f_\theta^{\mathcal{A}_t}$ is parameterized by $\theta$ of a target neural architecture $\mathcal{A}_t$. $C_s^{\mathcal{A}_s}$ outputs class probabilities by softmax function and $G_s$ is pre-trained on a labeled source dataset $\mathcal{D}_s$. We mainly consider classification problems and denote $f_\theta^{\mathcal{A}_t}$ as the target classifier $C_t^{\mathcal{A}_t}$ in the below.

In this setting, we assume the following conditions.

(i) **no label overlap**: $\mathcal{Y}_s \cap \mathcal{Y}_t = \emptyset$

(ii) **no source dataset access**: $\mathcal{D}_s$ is not available when training $C_t^{\mathcal{A}_t}$

(iii) **architecture inconsistency**: $\mathcal{A}_s \neq \mathcal{A}_t$

Existing methods are not available when the three conditions are satisfied simultaneously. Unsupervised domain adaptaion (Ganin et al., 2016) require $\mathcal{Y}_s \cap \mathcal{Y}_t \neq \emptyset$, accessing $\mathcal{D}_s$, and $\mathcal{A}_s = \mathcal{A}_t$. Source-free domain adaptation methods (Liang et al., 2020; Kundu et al., 2020; Wang et al., 2021a) can adapt models without accessing $\mathcal{D}_s$, but still depend on $\mathcal{Y}_s \cap \mathcal{Y}_t \neq \emptyset$ and $\mathcal{A}_s = \mathcal{A}_t$. Finetuning (Yosinski et al., 2014) can be applied to the problem with the condition (i) and (ii) if and only if $\mathcal{A}_s = \mathcal{A}_t$. However, since recent progress of neural architecture search (Zoph & Le, 2017; Wang et al., 2021b; Lee et al., 2021) enable to find specialized $\mathcal{A}_t$ for each task, situations of $\mathcal{A}_s \neq \mathcal{A}_t$ are common when developing models for environments requiring both accuracy and model size e.g., embedded devices. As a result, the specialized $C_t^{\mathcal{A}_t}$ currently sacrifices the accuracy so that the size requirement can be satisfied. Therefore, tackling this problem setting has the potential to enlarge the applicability of deep models.

## 3 PROPOSED METHOD

In this section, we describe our proposed method. An overview of our method is illustrated in Figure 1. PP yields initial weights for a target architecture by training it on the source task with synthesized samples from a conditional source generative model. P-SSL takes into account the pseudo samples drawn from generative models as an unlabeled dataset in an SSL setting. In P-SSL, we generate target-related samples by *pseudo conditional sampling* (PCS, Figure 2), which cascades a source classifier and a source conditional generative model.

### 3.1 PSEUDO PRE-TRAINING

Without accessing source datasets and architectures consistency, we cannot directly use the existing pre-trained weights for fine-tuning $C_t^{\mathcal{A}_t}$. To build useful representations under these conditions, we pre-train the weights of $\mathcal{A}_t$ with synthesized samples from a conditional source generative model. Every training iteration of PP is composed of two simple steps: (step 1) synthesizing a batch of generating source conditional samples $\{\hat{x}_s^i \sim G_s(y_s^i)\}_{i=1}^{B_s'}$ from uniformly sampled source labels $y_s^i \in \mathcal{Y}_s$, and (step 2) optimizing $\theta$ on the source classification task with the labeled batch of $\{(\hat{x}_s^i, y_s^i)\}_{i=1}^{B_s'}$ by minimizing

$$\frac{1}{B_s'} \sum_{i=1}^{B_s'} \mathrm{CE}(C_s^{\mathcal{A}_t}(\hat{x}_s^i; \theta), y_s^i), \tag{1}$$

where $B_s'$ is the batch size for PP and CE is cross-entropy loss. Since PP alternately performs the sample synthesis and training in an online manner, it efficiently yields pre-trained weights without consuming massive storage. Further, we found that this online strategy is better in accuracy than the offline strategy: i.e., synthesizing fixed samples in advance of training (Appendix C.4). We use the pre-trained weights from PP as the initial weights of $C_t^{\mathcal{A}_t}$ by replacing the final layer of the source task to that of the target task.

### 3.2 PSEUDO SEMI-SUPERVISED LEARNING

#### 3.2.1 SEMI-SUPERVISED LEARNING

Given a labeled dataset $\mathcal{D}_l = \{(x^i, y^i)\}_{i=1}^{N_l}$ and an unlabeled dataset $\mathcal{D}_u = \{(x^i)\}_{i=1}^{N_u}$, SSL is used to optimize the parameter $\theta$ of a deep neural network by solving the following problem.

$$\min_\theta \frac{1}{N_l} \sum_{(x,y)\in\mathcal{D}_l} \mathcal{L}_{\mathrm{sup}}(x, y, \theta) + \lambda \frac{1}{N_u} \sum_{x\in\mathcal{D}_u} \mathcal{L}_{\mathrm{unsup}}(x, \theta), \tag{2}$$

where $\mathcal{L}_{\mathrm{sup}}$ is a supervised loss for a labeled sample $(x_l, y_l)$ (e.g., cross-entropy loss), $\mathcal{L}_{\mathrm{unsup}}$ is an unsupervised loss for an unlabeled sample $x_u$, and $\lambda$ is a hyperparameter for balancing $\mathcal{L}_{\mathrm{sup}}$ and $\mathcal{L}_{\mathrm{unsup}}$. In SSL, it is generally assumed that $\mathcal{D}_l$ and $\mathcal{D}_u$ shares the same generative distribution $p(x)$. If there is a large gap between the labeled and unlabeled data distribution, the performance of SSL algorithms degrades (Oliver et al., 2018). However, Xie et al. (2020) have revealed that unlabeled samples in another dataset different from a target dataset can improve the performance of SSL algorithms by carefully selecting target-related samples from source datasets. This implies that SSL algorithms can achieve high performances as long as the unlabeled samples are related to target datasets, even when they belong to different datasets. On the basis of this implication, our P-SSL exploits pseudo samples drawn from source generative models as unlabeled data for SSL.

#### 3.2.2 PSEUDO CONDITIONAL SAMPLING

To generate informative target-related samples, our method uses PCS, which generates target-related samples by cascading $C_s^{\mathcal{A}_s}$ and $G_s$. With PCS, we first obtain a pseudo source label $y_{s\leftarrow t}$ from a source classifier $C_s^{\mathcal{A}_s}$ with a uniformly sampled $x_t$ from $\mathcal{D}_t$.

$$y_{s\leftarrow t} = C_s^{\mathcal{A}_s}(x_t) \tag{3}$$

Intuitively, $y_{s\leftarrow t}$ represents the relation between source class categories and $x_t$ in the form of the probabilities. We then generate target-related samples $x_{s\leftarrow t}$ with $y_{s\leftarrow t}$ as the conditional label by

$$x_{s\leftarrow t} \sim G_s(y_{s\leftarrow t}) = G_s(C_s^{\mathcal{A}_s}(x_t)). \tag{4}$$

**Algorithm 1** Pseudo conditional sampling

**Require:** Target dataset $\mathcal{D}_t$, source classifier $C_s^{\mathcal{A}_s}$, source generator $G_s$, number of pseudo samples $N_{s\leftarrow t}$, output label function $g$
**Ensure:** Pseudo unlabeled dataset $\mathcal{D}_{s\leftarrow t}$
1: $Y_{s\leftarrow t} \leftarrow \emptyset$
2: $\hat{C}_s^{\mathcal{A}_t} \leftarrow \text{SwapFinalLayer}(C_s^{\mathcal{A}_s}, l)$
3: **for** $x_t$ in $\mathcal{D}_t$ **do**
4: $\quad y_{s\leftarrow t} \leftarrow \hat{C}_s^{\mathcal{A}_s}(x_t)$
5: $\quad$ Add $y_{s\leftarrow t}$ to $Y_{s\leftarrow t}$
6: **end for**
7: Repeat concatenating $Y_{s\leftarrow t}$ with itself until the length reaches to $N_{s\leftarrow t}$
8: $\mathcal{D}_{s\leftarrow t} \leftarrow \emptyset$
9: **for** $y_{s\leftarrow t}$ in $Y_{s\leftarrow t}$ **do**
10: $\quad x_{s\leftarrow t} \sim G_s(y_{s\leftarrow t})$
11: $\quad$ Add $x_{s\leftarrow t}$ to $\mathcal{D}_{s\leftarrow t}$
12: **end for**

Table 2: Distribution gaps of pseudo samples on multiple target datasets

| | Distribution Gap (FID) | | |
|---|---|---|---|
| | $(\mathcal{D}_s, \mathcal{D}_t)$ | $(F(\mathcal{D}_s), \mathcal{D}_t)$ | $(\mathcal{D}_{s\leftarrow t}, \mathcal{D}_t)$ |
| Caltech-256-60 | 31.27 | 87.18 | 19.69 |
| CUB-200-2011 | 131.85 | 65.34 | 19.15 |
| DTD | 100.51 | 82.63 | 87.57 |
| FGVC-Aircraft | 189.16 | 47.29 | 23.68 |
| Indoor67 | 96.68 | 44.09 | 34.27 |
| OxfordFlower102 | 190.33 | 137.64 | 118.39 |
| OxfordPets | 95.16 | 20.59 | 16.94 |
| StanfordCars | 147.27 | 54.76 | 19.92 |
| StanfordDogs | 80.94 | 6.09 | 7.34 |

Although $G_s$ is trained with discrete (one-hot) class labels, it can generate class-wise interpolated samples by the continuously mixed labels of multiple class categories (Miyato & Koyama, 2018; Brock et al., 2019). By leveraging this characteristic, we aim to generate target-related samples by $y_{s\leftarrow t}$ constructed with an interpolation of source classes.

For the training of the target task, we compose a pseudo dataset $\mathcal{D}_{s\leftarrow t}$ by applying Algorithm 1. In line 2, we swap the final layer of $C_s^{\mathcal{A}_s}$ with an output label function $g$, which is softmax function as the default. Sec. C.5.3, we empirically evaluate the effects of the choice of $g$.

### 3.2.3 TRAINING

By applying PCS, we obtain a target-related sample $x_{s\leftarrow t}$ from $x_t$. In the training of $C_t^{\mathcal{A}_t}$, one can assign the label $y_t$ of $x_t$ to $x_{s\leftarrow t}$ since $x_{s\leftarrow t}$ is generated from $x_t$. However, it is difficult to directly use $(x_{s\leftarrow t}, y_t)$ in supervised learning because $x_{s\leftarrow t}$ can harm the performance of $C_t^{\mathcal{A}_t}$ due to the gap between label spaces; we empirically confirm that this naïve approach fails to boost the target performance mentioned in Sec. C.5.1. To extract informative knowledge from $x_{s\leftarrow t}$, we apply SSL algorithms that train a target classifier $C_t^{\mathcal{A}_t}$ by using a labeled target dataset $\mathcal{D}_t$ and an unlabeled dataset $\mathcal{D}_{s\leftarrow t}$ generated by PCS. On the basis of the implication discussed in Sec. 3.2.1, we can expect that the training by SSL improve $C_t^{\mathcal{A}_t}$ if $\mathcal{D}_{s\leftarrow t}$ contains target-related samples. We compute the unsupervised loss function for a pseudo sample $x_{s\leftarrow t}$ as $\mathcal{L}_{\text{unsup}}(x_{s\leftarrow t}, \theta)$. We can adopt arbitrary SSL algorithms for calculating $\mathcal{L}_{\text{unsup}}$. For instance, UDA (Xie et al., 2020) is defined by

$$\mathcal{L}_{\text{unsup}}(x, \theta) = \mathbb{1}\left(\max_{y' \in \hat{C}_t^{\mathcal{A}_t}(x, \tau); \theta} y' > \beta\right) \text{CE}\left(\hat{C}_t^{\mathcal{A}_t}(x, \tau; \theta), C_t^{\mathcal{A}_t}(T(x); \theta)\right), \quad (5)$$

where $\mathbb{1}$ is an indicator function, CE is a cross entropy function, $\hat{C}_t^{\mathcal{A}_t}(x, \tau)$ is the target classifier replacing the final layer with the temperature softmax function with a temperature hyperparameter $\tau$, $\beta$ is a confidence threshold, and $T(\cdot)$ is an input transformation function such as RandAugment (Cubuk et al., 2020). In the experiments, we used UDA because it achieves the best result; we compare and discuss applying the other SSL algorithms in Sec. C.5.1. Eventually, we optimize the parameter $\theta$ by the following objective function based on Eq (2).

$$\min_\theta \frac{1}{N_t} \sum_{x_t, y_t \in \mathcal{D}_t} \mathcal{L}_{\text{sup}}(x_t, y_t, \theta) + \lambda \frac{1}{N_{s\leftarrow t}} \sum_{x_{s\leftarrow t} \in \mathcal{D}_{s\leftarrow t}} \mathcal{L}_{\text{unsup}}(x_{s\leftarrow t}, \theta). \quad (6)$$

### 3.2.4 QUALITY OF PSEUDO SAMPLES

In this method, since we treat $x_{s\leftarrow t} \sim G_s(C_s^{\mathcal{A}_s}(x_t))$ as the unlabeled sample of the target task, the following assumption is required:

**Assumption 3.1** $p_{\mathcal{D}_t}(x) \approx p_{\mathcal{D}_{s\leftarrow t}}(x) = \frac{1}{N_t} \sum_{x_t} p_{G_s(C_s^{\mathcal{A}_s}(x_t))}(x)$,

where $p_{\mathcal{D}}(x)$ is a data distribution of a dataset $\mathcal{D}$. That is, if pseudo samples satisfy Assumption 3.1, P-SSL should boost the target task performance by P-SSL.

Table 3: List of target datasets

| Dataset | Task | Classes | Size (Train/Test) |
|---|---|---|---|
| Caltech-256-60 | General Object Recognition | 256 | 15,360/15,189 |
| CUB-200-2011 | Finegrained Object Recognition | 200 | 5,994/5,794 |
| DTD | Texture Recognition | 47 | 3,760/1,880 |
| FGVC-Aircraft | Finegrained Object Recognition | 100 | 6,667/3,333 |
| Indoor67 | Scene Recognition | 67 | 5,360/1,340 |
| OxfordFlower | Finegrained Object Recognition | 102 | 2,040/6,149 |
| OxfordPets | Finegrained Object Recognition | 37 | 3,680/3,369 |
| StanfordCars | Finegrained Object Recognition | 196 | 8,144/8,041 |

Table 4: Evaluations on Motivating Example

| | Architecture | Top-1 Acc. (%) |
|---|---|---|
| Scratch | RN18 $(4, 4, 4, 4)$ | $73.27^{\pm 0.43}$ |
| Fine-tuning | RN18 $(4, 4, 4, 4)$ | $87.75^{\pm 0.35}$ |
| Scratch | Custom RN18 $(2, 10, 2, 2)$ | $77.01^{\pm 0.57}$ |
| Logit Matching | Custom RN18 $(2, 10, 2, 2)$ | $85.81^{\pm 0.37}$ |
| Soft Target | Custom RN18 $(2, 10, 2, 2)$ | $82.67^{\pm 0.30}$ |
| Ours | Custom RN18 $(2, 10, 2, 2)$ | $\mathbf{89.13^{\pm 0.23}}$ |

To confirm that pseudo samples can satisfy Assumption 3.1, we assess the difference between the target distribution and pseudo distribution. Since it is difficult to directly compute the likelihood of the pseudo samples, we leverage Fréchet Inception Distance (FID, Heusel et al. (2017)), which measures a distribution gap between two datasets by 2-Wasserstein distance in the closed-form (lower FID means higher similarity). We evaluate the quality of $\mathcal{D}_{s \leftarrow t}$ by comparing FID($\mathcal{D}_{s \leftarrow t}, \mathcal{D}_t$) to FID($\mathcal{D}_s, \mathcal{D}_t$) and FID($F(\mathcal{D}_s), \mathcal{D}_t$), where $F(\mathcal{D}_s)$ is a subset of $\mathcal{D}_s$ constructed by confidence-based filtering similar to a previous study (Xie et al., 2020) (see Appendix B.4 for more detailed protocol). That is, if $\mathcal{D}_{s \leftarrow t}$ achieves lower FID than $\mathcal{D}_s$ or $\mathcal{D}'_s$, then $\mathcal{D}_{s \leftarrow t}$ can approximate $\mathcal{D}_t$ well.

Table 2 shows the FID scores when $\mathcal{D}_s$ is ImageNet. The experimental settings are shared with Sec. 4.4. Except for DTD and StanfordDogs, $\mathcal{D}_{s \leftarrow t}$ outperformed $\mathcal{D}_s$ and $F(\mathcal{D}_s)$ in terms of the similarity to $\mathcal{D}_t$. This indicates that PCS can produce more target-related samples than the natural source dataset. On the other hand, in the case of DTD (texture classes), $\mathcal{D}_{s \leftarrow t}$ relatively has low similarity to $\mathcal{D}_t$. This implies that PCS does not approximate $p_{\mathcal{D}_t}(x)$ well when the source and target datasets are not so relevant. Note that StanfordDogs is a subset of ImageNet, and thus FID($F(\mathcal{D}_s), \mathcal{D}_t$) was much smaller than other datasets. Nevertheless, the FID($\mathcal{D}_{s \leftarrow t}, \mathcal{D}_t$) is comparable to FID($F(\mathcal{D}_s), \mathcal{D}_t$). From this result, we can say that our method approximates the target distribution almost as well as the actual target samples. We further discuss the relationships between the pseudo sample quality and target performance in Sec. 4.4.

## 4 EXPERIMENTS

We evaluate our method with multiple target architectures and datasets, and compare it with baselines including scratch training and knowledge distillation that can be applied to our problem setting with a simple modification. We further conduct detailed analyses of our pseudo pre-training (PP) and pseudo semi-supervised learning (P-SSL) in terms of (a) the practicality of our method by comparing to the transfer learning methods that require the assumptions of source dataset access and architecture consistency, (b) the effect of source dataset choices, (c) the applicability toward another target task (object detection) other than classification. We further provide more detailed experiments including the effect of varying target dataset size (Appendix C.2), the performance difference when varying source generative model (Appendix C.3), the detailed analysis of PP (Appendix C.4) and P-SSL (Appendix C.5), and the qualitative evaluations of pseudo samples (Appendix D). We provide the detailed settings for training in Appendix B.3.

### 4.1 SETTING

**Baselines** Basically, there are no existing transfer learning methods that are available on the problem setting defined in Sec. 2. Thus, we evaluated our method by comparing it with the scratch training (**Scratch**), which trains a model with only a target dataset, and naïve applications of knowledge distillation methods: **Logit Matching** (Ba & Caruana, 2014) and **Soft Target** (Hinton et al., 2015). Logit Matching and Soft Target can be used for transfer learning under architecture inconsistency since their loss functions use only final logit outputs of models regardless of the intermediate layers. To transfer knowledge in $C_s^{\mathcal{A}_s}$ to $C_t^{\mathcal{A}_t}$, we first yield $C_t^{\mathcal{A}_s}$ by fine-tuning the parameters of $C_s^{\mathcal{A}_s}$ on the target task, and then train $C_t^{\mathcal{A}_t}$ with knowledge distillation penalties by treating $C_t^{\mathcal{A}_s}$ as the teacher model. We provide more details in Appendix B.2.

**Datasets** We used ImageNet (Russakovsky et al., 2015) as the default source datasets. In Sec. 4.6, we report the case of applying CompCars (Yang et al., 2015) as the source dataset. For the target dataset, we mainly used StanfordCars (Krause et al., 2013), which is for fine-grained classification of car types, as the target dataset. In Sec. 4.4, we used the nine classification datasets listed in

Table 5: Performance comparison of multiple target architectures on StanfordCars (Top-1 Acc.(%))

| | ResNet-50 | WRN-50-2 | MNASNet1.0 | MobileNetV3-L | EfficientNet-B0 | EfficientNet-B5 |
|---|---|---|---|---|---|---|
| Scratch | $71.86^{\pm0.80}$ | $76.21^{\pm1.40}$ | $79.22^{\pm0.66}$ | $80.98^{\pm0.27}$ | $80.80^{\pm0.56}$ | $81.73^{\pm3.50}$ |
| Logit Matching | $84.36^{\pm0.47}$ | $86.28^{\pm1.13}$ | $85.08^{\pm0.08}$ | $85.11^{\pm0.10}$ | $86.37^{\pm0.69}$ | $88.42^{\pm0.60}$ |
| Soft Target | $79.95^{\pm1.62}$ | $82.34^{\pm1.15}$ | $83.55^{\pm1.21}$ | $84.64^{\pm0.21}$ | $85.16^{\pm0.69}$ | $85.30^{\pm1.37}$ |
| Ours | $\mathbf{90.69^{\pm0.11}}$ | $\mathbf{91.76^{\pm0.41}}$ | $\mathbf{87.39^{\pm0.19}}$ | $\mathbf{88.40^{\pm0.67}}$ | $\mathbf{89.28^{\pm0.41}}$ | $\mathbf{90.04^{\pm0.34}}$ |

Table 6: Performance comparison of WRN-50-2 classifiers on multiple target datasets (Top-1 Acc. (%))

| | Caltech-256-60 | CUB-200-2011 | DTD | FGVC-Aircraft | Indoor67 | OxfordFlower | OxfordPets | StanfordDogs |
|---|---|---|---|---|---|---|---|---|
| Scratch | $48.07^{\pm1.30}$ | $52.61^{\pm1.36}$ | $45.11^{\pm2.37}$ | $74.04^{\pm0.59}$ | $50.77^{\pm1.07}$ | $67.91^{\pm0.94}$ | $63.03^{\pm1.59}$ | $57.16^{\pm3.11}$ |
| Logit Matching | $55.28^{\pm2.63}$ | $62.52^{\pm2.13}$ | $49.29^{\pm0.69}$ | $78.91^{\pm1.93}$ | $57.61^{\pm0.74}$ | $75.23^{\pm1.11}$ | $70.56^{\pm3.96}$ | $64.46^{\pm1.88}$ |
| Soft Target | $54.84^{\pm1.33}$ | $60.53^{\pm1.65}$ | $48.39^{\pm1.06}$ | $77.08^{\pm3.30}$ | $54.08^{\pm1.22}$ | $69.90^{\pm0.38}$ | $65.62^{\pm0.97}$ | $63.64^{\pm3.00}$ |
| Ours w/o PP | $51.62^{\pm0.79}$ | $56.61^{\pm1.31}$ | $45.50^{\pm0.17}$ | $77.13^{\pm0.46}$ | $51.23^{\pm1.01}$ | $68.11^{\pm1.21}$ | $68.70^{\pm1.21}$ | $61.26^{\pm0.85}$ |
| Ours w/o P-SSL | $70.88^{\pm0.21}$ | $71.78^{\pm0.28}$ | $\mathbf{61.28^{\pm0.66}}$ | $86.08^{\pm0.14}$ | $66.79^{\pm0.22}$ | $\mathbf{94.02^{\pm0.27}}$ | $86.31^{\pm0.10}$ | $73.30^{\pm0.10}$ |
| Ours | $\mathbf{71.35^{\pm0.32}}$ | $\mathbf{74.93^{\pm0.16}}$ | $57.48^{\pm1.28}$ | $\mathbf{87.98^{\pm0.91}}$ | $\mathbf{67.72^{\pm0.11}}$ | $90.31^{\pm0.17}$ | $\mathbf{89.97^{\pm0.41}}$ | $\mathbf{75.25^{\pm0.13}}$ |

Table 3 as the target datasets. We used these datasets with the intent to include various granularities and domains. We constructed Caltech-256-60 by randomly sampling 60 images per class from the original dataset in accordance with the procedure of Cui et al. (2018). Note that StanfordDogs is a subset of ImageNet, and thus has the overlap of the label space to ImageNet, but we added this dataset to confirm the performance when the overlapping exists.

**Network Architecture** As a source architecture $\mathcal{A}_s$, we used the ResNet-50 architecture (He et al., 2016) with the pre-trained weight distributed by `torchvision` official repository.[1] For a target architecture $\mathcal{A}_t$, we used five architectures publicly available on `torchvision`: WRN-50-2 (Zagoruyko & Komodakis, 2016), MNASNet1.0 (Tan et al., 2019), MobileNetV3-L (Howard et al., 2019), and EfficientNet-B0/B5 (Tan & Le, 2019). Note that, to ensure reproducibility, we assume them as entirely new architectures for the target task in our problem setting, while they are actually existing architectures. For a source conditional generative model $G_s$, we used BigGAN (Brock et al., 2018) generating $256 \times 256$ resolution images as the default architecture. We also tested the other generative models such as ADM-G (Dhariwal & Nichol, 2021) in Sec. C.3. We implemented BigGAN on the basis of open source repositories including `pytorch-pretrained-BigGAN`[2]; we used the pre-trained weights distributed by the repositories.

## 4.2 MOTIVATING EXAMPLE: MANUAL ARCHITECTURE SEARCH

First of all, we confirm the effectiveness of our setting and method through a practical scenario described in Sec. 1. Here, we consider a case of manually optimizing the number of layers of ResNet-18 (RN18) for the target task to improve the performance while keeping the model size.

We evaluated our method on the above scenario by assuming StanfordCars as the target dataset and ImageNet as the source dataset. We searched the custom architecture by grid search of the layers in four blocks of RN18 from $(2,2,2,2)$ to $(2,2,2,10)$ by varying layers in $\{2,4,6,8,10\}$ for each block of ResNet while keeping the sum of layers less than or equal to 18 to maintain architecture size. We found that the best architecture is one with $(2,10,2,2)$. The test accuracies on StanfordCars are shown in Table 4. We can see that finding an optimal custom RN18 architecture for the target task brings test accuracy improvements, and our method contributes to further improvements under this difficult situation. This result indicates that our method can widen the applicability of neural architecture search techniques, which have been difficult in practice in terms of accuracy.

## 4.3 TARGET ARCHITECTURES

We discuss the performance evaluations by varying the neural network architectures of the target classifiers to evaluate our method on the condition of architecture inconsistency. Table 5 lists the results on StanfordCars with multiple target architectures. Note that we evaluated our method by fixing the source architecture to ResNet-50. Our method outperformed the baselines on all target architectures without architecture consistency. Remarkably, our method stably performs arbitrary

---

[1]https://github.com/pytorch/vision
[2]https://github.com/huggingface/pytorch-pretrained-BigGAN

Table 7: Comparison to fine-tuning (FT) and semi-supervised learning (SSL) on StanfordCars

| | Top-1 Acc. (%) | |
| --- | --- | --- |
| **Generative Model** | PP | P-SSL |
| BigGAN (Brock et al., 2019) | $\mathbf{90.95}^{\pm 0.21}$ | $\mathbf{80.01}^{\pm 0.14}$ |
| **Real Dataset** | FT | R-SSL |
| All ImageNet | $88.24^{\pm 0.17}$ | $73.01^{\pm 3.10}$ |
| Filtered ImageNet | $74.08^{\pm 2.77}$ | $77.21^{\pm 0.29}$ |

Table 8: Comparison of source datasets on StanfordCars

| | Source Dataset | |
| --- | --- | --- |
| | ImageNet | CompCars |
| PP | $90.95^{\pm 0.21}$ | $87.57^{\pm 0.32}$ |
| P-SSL | $80.01^{\pm 0.14}$ | $79.97^{\pm 0.13}$ |
| PP + P-SSL | $91.76^{\pm 0.41}$ | $87.80^{\pm 0.33}$ |

relationships between target and source architectures including from a smaller architecture (ResNet-50) to larger ones (WRN-50-2 and EfficientNet-B5), and from a larger one (ResNet-50) to smaller ones (MNASNet1.0, MobileNetV3, and EfficientNet-B0). This flexibility can lead to the effectiveness of the neural architecture search in Sec. 4.2.

## 4.4 TARGET DATASETS

We show the efficacy of our method on multiple target datasets other than StanfordCars. We used WRN-50-2 as the architecture of the target classifiers. Table 6 lists the top-1 accuracy of each classification task. Our method stably improved the baselines across datasets. We also print the ablation results of our method (Ours w/o PP and Ours w/o P-UDA) for assessing the dependencies of PP and P-SSL on datasets. PP outperformed the baselines on all target datasets. This suggests that building basic representation by pre-training is effective for various target tasks even if the source dataset is drawn from generative models. For P-SSL, we observed that it boosted the scratch models in all target datasets except for DTD and OxfordFlower. As the reasons for the degradation on DTD and OxfordFlower, we consider that the pseudo samples do not satisfy Assumption 3.1 as discussed in Sec. 3.2.4. In fact,

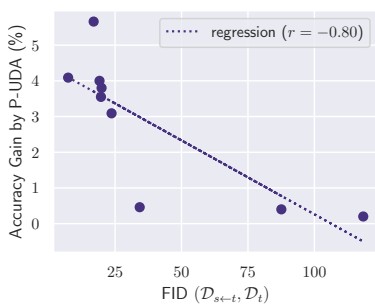

Figure 3: Correlation between FID and accuracy gain

we observe that $\text{FID}(\mathcal{D}_{s \leftarrow t}, \mathcal{D}_t)$ is correlated to the accuracy gain from Scratch models ($-0.80$ of correlation coefficient, $-0.97$ of Spearman rank correlation) as shown in Fig. 3. These experimental results suggest that our method is effective on the setting without no label overlap as long as $\mathcal{D}_{s \leftarrow t}$ approximates $\mathcal{D}_t$ well.

## 4.5 COMPARISON TO METHODS REQUIRING PARTIAL CONDITIONS

To confirm the practicality of our method, we compared it with methods requiring $\mathcal{A}_s = \mathcal{A}_t$ and accessing $\mathcal{D}_s$. We tested **Fine-tuning (FT)** and **R-SSL**, which use a source pre-trained model and a real source dataset for SSL as reference. For FT and R-SSL, we used ImageNet and a subset of ImageNet (Filtered ImageNet), which was collected by confidence-based filtering similar to Xie et al. (2020) (see Appendix B.4). This manual filtering process corresponds to PCS in P-SSL. In Table 7, PP and P-SSL outperformed FT and R-SSL, respectively. This suggests that the samples from $G_s$ not only preserve essential information of $\mathcal{D}_s$ but are also more useful than $\mathcal{D}_s$, i.e., accessing $\mathcal{D}_s$ may not be necessary for transfer learning. In summary, PP and P-SSL are practical enough compared to existing methods that require assumptions.

## 4.6 SOURCE DATASETS

We investigate the preferable characteristics of source datasets for PP and P-SSL by testing another source dataset, which was CompCars (Yang et al., 2015), a fine-grained vehicle dataset containing 136K images (see Appendix B.5 for details). This is more similar to the target dataset (StanfordCars) than ImageNet. All training settings were the same as mentioned in Sec. 4.1. Table 8 lists the scores for each model of our methods. The models using ImageNet were superior to those using CompCars. To seek the difference, we measured $\text{FID}(\mathcal{D}_{s \leftarrow t}, \mathcal{D}_t)$ when using CompCars as with the protocol in Sec. 4.5, and the score was 22.12, which is inferior to 19.92 when using ImageNet. This suggests that the fidelity of pseudo samples toward target samples is important to boost the target performance and is not simply determined by the similarity between the source and target datasets. We consider

that the diversity of source classes provides the usefulness of pseudo samples; thus ImageNet is superior to CompCars as the source dataset.

### 4.7 APPLICATION TO OBJECT DETECTION TASK

Although we have mainly discussed the cases where the target task is classification through the paper, our method can be applied to any task for which the SSL method exists. Here, we evaluate the applicability of PP+P-SSL toward another target task other than classification. To this end, we applied our method to object detection task on PASCAL-VOC 2007 (Everingham

Table 9: Results on PASCAL-VOC 2007

|  | trainval07+12 | | | trainval07 | | |
|---|---|---|---|---|---|---|
|  | AP | $AP_{50}$ | $AP_{75}$ | AP | $AP_{50}$ | $AP_{75}$ |
| FT (ImageNet) | 52.5 | 80.3 | 56.2 | 42.2 | 74.0 | 43.7 |
| PP | 52.6 | 80.1 | 57.4 | 41.2 | 70.6 | 43.4 |
| PP+P-SSL (UT) | **57.3** | **83.1** | **63.5** | **52.2** | **80.3** | **56.8** |

et al., 2015) by using FPN (Lin et al., 2017) models with ResNet-50 backbone. As the SSL method, we used Unbiased Teacher (UT, (Liu et al., 2021)), which is a method for object detection based on self distillation and pseudo labeling, and implemented PP+P-SSL on the code base provided by Liu et al. (2021). We generated samples for PP and P-SSL in the same way as the classification experiments; we used ImageNet as the source dataset. Table 9 shows the results of average precision scores calculated by following detectron2 (Wu et al., 2019). Note that the baseline of Table 9 is FT instead of Scratch because the Scratch setting of the object detection task is hard to train and too slow to converge. We confirm that PP achieved competitive results to FT and P-SSL significantly boosted the PP model. This can be caused by the high similarity between $\mathcal{D}_{s \leftarrow t}$ and $\mathcal{D}_t$ (19.0 in FID). This result indicates that our method has the flexibility to be applied to other tasks as well as classification and is expected to improve baselines.

## 5 RELATED WORK

We briefly discuss related works by focusing on training techniques applying generative models. We also provide discussions on existing transfer learning and semi-supervised learning in Appendix A.

Similar to our study, several studies have applied the expressive power of conditional generative models to boost the performance of discriminative models; Zhu et al. (2018) and Yamaguchi et al. (2020) have exploited the generated images from conditional GANs for data augmentation in classification, and Sankaranarayanan et al. (2018) have introduced conditional GANs into the system of domain adaptation for learning joint-feature spaces of source and target domains. Moreover, Li et al. (2020b) have implemented an unsupervised domain adaptation technique with conditional GANs in a setting of no accessing source datasets. These studies require label overlapping between source and target tasks or training of generative models on target datasets, which causes problems of overfitting and mode collapse when the target datasets are small (Zhang et al., 2020; Zhao et al., 2020b; Karras et al., 2020). Our method, however, requires no additional training in generative models because it simply extracts samples from fixed pre-trained conditional generative models.

## 6 CONCLUSION AND LIMITATION

We explored a new transfer learning setting where (i) source and target task label spaces do not have overlaps, (ii) source datasets are not available, and (iii) target network architectures are not consistent with source ones. In this setting, we cannot use existing transfer learning such as domain adaptation and fine-tuning. To transfer knowledge, we proposed a simple method leveraging pre-trained conditional source generative models, which is composed of PP and P-SSL. PP yields effective initial weights of a target architecture by generated samples and P-SSL applies an SSL algorithm by taking into account the pseudo samples from the generative models as the unlabeled dataset for the target task. Our experiments showed that our method can practically transfer knowledge from the source task to the target tasks without the assumptions of existing transfer learning settings. One of the limitations of our method is the difficulty to improve target models when the gap between source and target is too large. A future step is to modify the pseudo sampling process by optimizing generative models toward the target dataset, which was avoided in this work to keep the simplicity and stability of the method.

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

APPENDIX

The following manuscript provides the supplementary materials of the main paper: Transfer Learning with Pre-trained Conditional Generative Models. We describe (A) additional related works of domain adaptation and fine-tuning, (B) details of experimental settings used in the main paper, (C) additional experiments including comparison of our method and fine-tuning, and detailed analysis of PP and P-SSL, and (D) qualitative evaluations of pseudo samples by PCS.

# A  EXTENDED RELATED WORK

## A.1  DOMAIN ADAPTATION

Domain adaptation leverages source knowledge to the target task by minimizing domain gaps between source and target domains through joint-training (Ganin et al., 2016). It is generally assumed that the source and target task label spaces overlap (Pan & Yang, 2010; Wang & Deng, 2018) and labeled source datasets are available when training target models. Several studies have attempted to solve the transfer learning problems called source-free adaptation (Chidlovskii et al., 2016; Liang et al., 2020; Kundu et al., 2020; Wang et al., 2021a), where the model must adapt to the target domain without target labels and the source dataset. However, they still require architecture consistency and overlaps between the source and target tasks, i.e., it is not applicable to our problem setting.

## A.2  FINETUNING

Our method is categorized as an inductive transfer learning approach (Pan & Yang, 2010), where thelabeled target datasets are available and the source and target task label spaces do not overlap. In deep learning, fine-tuning (Yosinski et al., 2014; Agrawal et al., 2014; Girshick et al., 2014), which leverages source pre-trained weights as initial parameters of the target modesl, is one of the most common approaches of inductive transfer learning because of its simplicity. Previous studies have attempted to improve fine-tuning by adding a penalty of the gaps between source and target models such as adding $L^2$ penalty term (Li et al., 2018) or penalty using channel-wise importance of feature maps (Li et al., 2019). You et al. (2020) have introduced category relationships between source and target tasks into target-task training and penalized the target models to predict pseudo source labels that are the outputs of the source models by applying target data. Shu et al. (2021) have presented an approach leveraging multiple source models pre-trained on different datasets and tasks by mixing the outputs via adaptive aggregation modules. Although these methods outperform the naïve fine-tuning baselines, they require architecture consistency between source and target tasks. In contrast to the fine-tuning methods, our method can be used without architecture consistency and source dataset access since it transfers source knowledge via pseudo samples drawn from source pre-trained generative models.

## A.3  SEMI-SUPERVISED LEARNING

SSL is a paradigm that trains a supervised model with labeled and unlabeled samples by minimizing supervised and unsupervised loss simultaneously. Historically, various SSL algorithms have been used or proposed for deep learning such as entropy minimization (Grandvalet & Bengio, 2005), pseudo-label (Lee et al., 2013), virtual adversarial training (Miyato et al., 2017), and consistency regularization (Bachman et al., 2014; Sajjadi et al., 2016; Laine & Aila, 2016). UDA (Xie et al., 2020) and FixMatch (Sohn et al., 2020), which combine ideas of pseudo-label and consistency regularization, have achieved remarkable performance. An assumption with these SSL algorithms is that the unlabeled data are sampled from the same distribution as the labeled data. If there is a large gap between the labeled and unlabeled data distribution, the performance of SSL algorithms degrades (Oliver et al., 2018). However, Xie et al. (2020) have revealed that unlabeled samples in another dataset different from a target dataset can improve the performance of SSL algorithms by carefully selecting target-related samples from source datasets. This indicates that SSL algorithms can achieve high performances as long as the unlabeled samples are related to target datasets, even when they belong to different datasets. On the basis of this implication, our P-SSL exploits pseudo samples drawn from source generative models as unlabeled data for SSL. We tested SSL algorithms

for P-SSL and compared the resulting P-SSL models with SSL models using the filtered real source dataset constructed by the protocol of Xie et al. (2020) in Appendix C.5.1 and Sec. 4.5.

# B    DETAILS OF EXPERIMENTS

## B.1    DATASET DETAILS

**ImageNet** (Russakovsky et al., 2015): We downloaded ImageNet from the official site `https://www.image-net.org/`. ImageNet is released under license that allows it to be used for non-commercial research/educational purposes (see `https://image-net.org/download.php`).

**Caltech-256** (Griffin et al., 2007): We downloaded Caltech-256 from the official site `https://data.caltech.edu/records/20087`. Caltech-256 is released under CC-BY license.

**CUB-200-2011** (Welinder et al., 2010): We downloaded CUB-200-2011 from the official site `http://www.vision.caltech.edu/datasets/cub_200_2011/`. CUB-200-2011 is released under license that allows it to be used for non-commercial purposes (see `https://authors.library.caltech.edu/27452/`).

**DTD** (Cimpoi et al., 2014): We downloaded DTD from the official site `https://www.robots.ox.ac.uk/~vgg/data/dtd/`. DTD is released under license that allows it to be used for non-commercial research purposes (see `https://www.robots.ox.ac.uk/~vgg/data/dtd/`).

**FGVC-Aircraft** (Maji et al., 2013): We downloaded FGVC-Aircraft from the official site`https://www.robots.ox.ac.uk/~vgg/data/fgvc-aircraft/`. FGVC-Aircraft is released under license that allows it to be used for non-commercial research purposes (see `https://www.robots.ox.ac.uk/~vgg/data/fgvc-aircraft/`).

**Indoor67** (Quattoni & Torralba, 2009): We downloaded Indoor67 from the official site`https://web.mit.edu/torralba/www/indoor.html`. Indoor67 is released under license that allows it to be used for non-commercial research purposes (see `https://web.mit.edu/torralba/www/indoor.html`).

**OxfordFlower** (Nilsback & Zisserman, 2008): We downloaded OxfordFlower from the official site`https://www.robots.ox.ac.uk/~vgg/data/flowers/102/`. OxfordFlower is released under unknown license.

**OxfordPets** (Parkhi et al., 2012): We downloaded OxfordPets from the official site`https://www.robots.ox.ac.uk/~vgg/data/pets/`. OxfordPets is released under Creative Commons Attribution-ShareAlike 4.0 International License.

**StanfordCars** (Krause et al., 2013): We downloaded StanfordCars from the official site`https://ai.stanford.edu/~jkrause/cars/car_dataset.html`. StanfordCars is released under license that allows it to be used for non-commercial research purposes (see `https://ai.stanford.edu/~jkrause/cars/car_dataset.html`).

**StanfordDogs** (Khosla et al., 2011): We downloaded StanfordDogs from the official site`http://vision.stanford.edu/aditya86/ImageNetDogs/`. StanfordDogs is released under license that allows it to be used for non-commercial research/educational purposes (see `https://image-net.org/download.php`).

**CompCars** (Yang et al., 2015): We downloaded CompCars from the official site`http://mmlab.ie.cuhk.edu.hk/datasets/comp_cars/`. CompCars is released under license that allows it to be used for non-commercial research purposes (see `http://mmlab.ie.cuhk.edu.hk/datasets/comp_cars/`).

**PASCAL-VOC** (Everingham et al., 2015): We downloaded PASCAL-VOC from the official site`http://host.robots.ox.ac.uk/pascal/VOC/`. PASCAL-VOC is released under license of flickr (see `https://www.flickr.com/help/terms`).

## B.2 Knowledge Distillation Baselines

As stated in the main paper, we evaluated our method by comparing it with naïve knowledge distillation methods **Logit Matching** (Ba & Caruana, 2014) and **Soft Target** (Hinton et al., 2015). We exploited Logit Matching and Soft Target for transfer learning under the architecture inconsistency since their loss functions use only final logit outputs of models regardless of the intermediate layers. To tranfer knowledge in $C_s^{\mathcal{A}_s}$ to $C_t^{\mathcal{A}_t}$, we first fine-tune the parameters of $C_s^{\mathcal{A}_s}$ on the target task then train $C_t$ with knowledge distillation penalties by treating the trained $C_s^{\mathcal{A}_s}$ as the teacher model. We optimized the knowledge distillation models by the following objective function:

$$\min_{\theta} \sum_{x_t, y_t \in \mathcal{D}_t} \mathcal{L}_{\text{sup}}(x_t, y_t, \theta) + \lambda_d \mathcal{L}_{\text{KD}}(l_\theta(x_t), l_\phi(x_t)), \tag{7}$$

where $\lambda_d$ is a hyperparameter, $\mathcal{L}_{\text{KD}}$ is a loss function of a knowledge distillation method, $\phi$ is the parameter of a teacher model, and $l_\theta(\cdot)$ and $l_\phi(\cdot)$ are the output of the logit function on $\theta$ and $\phi$. In Logit Matching, $\mathcal{L}_{\text{KD}}$ is a simple mean squared loss between $l_\theta$ and $l_\phi$. Soft Target computes a Kullback-Leibler divergence between the softmax output of $l_\theta$ and the temperature softmax output of $l_\phi$ as $\mathcal{L}_{\text{KD}}$. We set the temperature parameter $T$ to 4 by searching in $\{2, 4, 6\}$.

## B.3 Training Settings

We selected the training configurations on the basis of the previous works (Li et al., 2020a; Xie et al., 2020). In PP, we trained a source classifier by Neterov momentum SGD for 1M iterations with a mini-batch size of 128, weight decay of 0.0001, momentum of 0.9, and initial learning rate of 0.1; we decayed the learning rate by 0.1 at 30, 60, 90 epochs. We trained a target classifier $C_t^{\mathcal{A}_t}$ by Neterov momentum SGD for 300 epochs with a mini-batch size of 16, weight decay of 0.0001, and momentum of 0.9. We set the initial learning rate to 0.05 for the scratch models and 0.005 for the models with PP. We dropped the learning rate by 0.1 at 150 and 250 epochs. For each target dataset, we split the training set into $9 : 1$, and used the former in the training and the later in validating. The input samples were resized into $224 \times 224$ resolution. For the SSL algorithms, we set the mini-batch size for $\mathcal{L}_{\text{unsup}}$ to 112, and fixed $\lambda$ in Eq. (2) to 1.0. We fixed the hyperparameters of UDA as the confidence threshold $\beta = 0.5$ and the temperature parameter $\tau = 0.4$ following Xie et al. (2020). For P-SSL, we generated 50,000 samples by PCS. We ran the target trainings three times with different seeds and selected the best models in terms of the validation accuracy for each epoch. We report the average top-1 test accuracies and standard deviations.

## B.4 Filtering of Real Dataset by Relation to Target

We provide the details of the protocol of dataset filtering discussed in Sec. 4.5 of the main paper and list the correspondences between the target classes and selected source classes. For filtering datasets, we first calculated the confidence (the maximum probability of a class in the prediction) of the target samples by using the pre-trained source classifiers, then averaged the confidence scores for each target class, and finally selected the source classes with a confidence higher than 0.001 as the unlabeled dataset similar to (Xie et al., 2020). We list the filtered ImageNet classes for each target dataset in Table 24 and 25.

## B.5 Source Dataset: CompCars

CompCars (Yang et al., 2015) is a fine-grained vehicle image dataset for classifying the vehicle manufacturers or models. It contains 163 manufacturer classes, 1,716 model classes, and 136,726 images of entire vehicles collected from the web. As the source dataset for PP and P-SSL, we used 163 manufacturer classes for training classifiers and conditional generative models since the manufacturer classes do not overlap with the classes of the target dataset, i.e., StanfordCars.

Table 10: Comparison of our methods with fine-tuning methods

|  | Top-1 Acc. (%) | |
| --- | --- | --- |
|  | Baseline | + P-SSL |
| Pseudo Pre-training (Ours) | $90.95^{\pm 0.21}$ | $91.76^{\pm 0.41}$ |
| Fine-tuning | $90.56^{\pm 0.17}$ | $91.41^{\pm 0.15}$ |
| L2-SP (Li et al., 2018) | $91.20^{\pm 0.05}$ | $91.43^{\pm 0.10}$ |
| DELTA (Li et al., 2019) | $91.52^{\pm 0.26}$ | $\mathbf{91.82^{\pm 0.29}}$ |
| BSS (Chen et al., 2019) | $91.25^{\pm 0.27}$ | $91.45^{\pm 0.16}$ |
| Co-Tuning (You et al., 2020) | $91.08^{\pm 0.15}$ | $91.16^{\pm 0.05}$ |

Table 11: Comparison of our methods with fine-tuning discriminator

|  | Top-1 Acc. (%) |
| --- | --- |
| Scratch (WRN-50-2) | $76.92^{\pm 2.16}$ |
| Finetuning Discriminator (BigGAN) | $16.09^{\pm 0.10}$ |
| Ours (WRN-50-2) | $91.76^{\pm 0.41}$ |

Table 12: Performance comparison of multiple target architectures on StanfordCars (Top-1 Acc. (%))

|  | ResNet-50 | WRN-50-2 | MNASNet1.0 | MobileNetV3-L | EfficientNet-B0 | EfficientNet-B5 |
| --- | --- | --- | --- | --- | --- | --- |
| Scratch | $71.86^{\pm 0.80}$ | $76.21^{\pm 1.40}$ | $79.22^{\pm 0.66}$ | $80.98^{\pm 0.27}$ | $80.80^{\pm 0.56}$ | $81.73^{\pm 3.50}$ |
| Logit Matching | $84.36^{\pm 0.47}$ | $86.28^{\pm 1.13}$ | $85.08^{\pm 0.08}$ | $85.11^{\pm 0.10}$ | $86.37^{\pm 0.69}$ | $88.42^{\pm 0.60}$ |
| Soft Target | $79.95^{\pm 1.62}$ | $82.34^{\pm 1.15}$ | $83.55^{\pm 1.21}$ | $84.64^{\pm 0.21}$ | $85.16^{\pm 0.69}$ | $85.30^{\pm 1.37}$ |
| Ours w/o PP | $80.14^{\pm 0.57}$ | $80.01^{\pm 0.14}$ | $82.22^{\pm 0.16}$ | $83.26^{\pm 0.21}$ | $83.22^{\pm 0.54}$ | $85.82^{\pm 0.47}$ |
| Ours w/o P-SSL | $90.25^{\pm 0.19}$ | $90.95^{\pm 0.21}$ | $87.14^{\pm 0.05}$ | $87.82^{\pm 0.83}$ | $88.27^{\pm 0.33}$ | $89.50^{\pm 0.17}$ |
| Ours | $\mathbf{90.69^{\pm 0.11}}$ | $\mathbf{91.76^{\pm 0.41}}$ | $\mathbf{87.39^{\pm 0.19}}$ | $\mathbf{88.40^{\pm 0.67}}$ | $\mathbf{89.28^{\pm 0.41}}$ | $\mathbf{90.04^{\pm 0.34}}$ |
| Fine-tuning (FT) | $90.56^{\pm 0.17}$ | $88.24^{\pm 1.55}$ | $89.18^{\pm 0.14}$ | $87.57^{\pm 0.28}$ | $90.06^{\pm 0.25}$ | $91.64^{\pm 0.29}$ |
| FT + P-SSL | $\mathbf{91.41^{\pm 0.15}}$ | $\mathbf{91.95^{\pm 0.09}}$ | $\mathbf{89.46^{\pm 0.15}}$ | $\mathbf{88.19^{\pm 0.15}}$ | $\mathbf{90.13^{\pm 0.14}}$ | $\mathbf{91.71^{\pm 0.09}}$ |
| R-SSL with ImageNet | $77.48^{\pm 0.46}$ | $77.93^{\pm 1.45}$ | $81.74^{\pm 2.19}$ | $82.17^{\pm 0.75}$ | $82.25^{\pm 1.03}$ | $83.86^{\pm 1.54}$ |
| FT + R-SSL | $90.70^{\pm 0.17}$ | $91.87^{\pm 0.22}$ | $88.63^{\pm 0.20}$ | $87.26^{\pm 0.09}$ | $89.91^{\pm 0.19}$ | $91.10^{\pm 0.08}$ |

## C  ADDITIONAL EXPERIMENTS

### C.1  COMPARISON OF OUR METHOD WITH FINE-TUNING METHODS

For assessing the practicality of our method, we additionally compare our method with the fine-tuning methods that require architecture consistency, i.e., **Fine-tuning**: naïvely training target classifiers by using the source pre-trained weights as the initial weights. **L2-SP** (Li et al., 2018): fine-tuning with the $L^2$ penalty term between the current training weights and the pre-trained source weights. **DELTA** (Li et al., 2019): fine-tuning with a penalty minimizing the gaps of channel-wise outputs of feature maps between source and target models. **BSS** (Chen et al., 2019): fine-tuning with the penalty term enlarging eigenvalues of training features to avoid negative transfer. **Co-Tuning** (You et al., 2020): fine-tuning on source and target task simultaneously by translating the target labels to the source labels. We implemented these methods on the basis of the open source repositories provided by the authors. All of hyperparameters used in L2-SP, DELTA, BSS, and Co-Tuning are those in the respective papers: $\beta$ for L2-SP and DELTA was 0.01, $\eta$ for BSS was 0.001, and $\lambda$ for Co-Tuning was 2.3. We also tested the models by combining our methods and the fine-tuning methods. Table 12 and 13 list the extended results of the experiments on multiple architectures and target datasets in Secs. 4.3 and 4.4, respectively. Table 10 summarizes the results using the fine-tuning variants. The +P-SSL column indicates the results of the combination models of a fine-tuning method and P-SSL. We confirm that our method (pseudo pre-training + P-SSL) achieved competitive or superior results to the naïve fine-tuning. This means that our method can improve target models as well as fine-tuning without architectures consistency. We also observed that P-SSL can outperform fine-tuning baselines by being combined with them. These results indicate that our P-SSL can be applied even if the source and target architectures are consistent.

### C.1.1  COMPARSION TO FINE-TUNING DISCRIMINATOR

We compare PP with a transfer learning method applying encoders of generative models as the pre-trained encoder for target tasks. Here, we assume the discriminator of ImageNet pre-trained BigGAN as the pre-trained encoder, and fine-tune it on the target classification task of StanfordCars. Table 11 shows the results. The BigGAN discriminator easily overfitted the target dataset and catastrophically degraded the test performance. It seems difficult to achieve high accuracy by naïvely applying the discriminator as a simple pre-trained encoder. This suggests that the representation of

Table 13: Performance comparison of classifiers on multiple target datasets (Top-1 Acc. (%))

| | Caltech-256-60 | CUB-200-2011 | DTD | FGVC-Aircraft | Indoor67 | OxfordFlower | OxfordPets | StanfordDogs |
|---|---|---|---|---|---|---|---|---|
| Scratch | $48.07^{\pm1.30}$ | $52.61^{\pm1.36}$ | $45.11^{\pm2.37}$ | $74.04^{\pm0.59}$ | $50.77^{\pm1.07}$ | $67.91^{\pm0.94}$ | $63.03^{\pm1.59}$ | $57.16^{\pm3.11}$ |
| Logit Matching | $55.28^{\pm2.63}$ | $62.52^{\pm2.13}$ | $49.29^{\pm0.69}$ | $78.91^{\pm1.93}$ | $57.61^{\pm0.74}$ | $75.23^{\pm1.11}$ | $70.56^{\pm3.96}$ | $64.46^{\pm1.88}$ |
| Soft Target | $54.84^{\pm1.33}$ | $60.53^{\pm1.65}$ | $48.39^{\pm1.06}$ | $77.08^{\pm3.30}$ | $54.08^{\pm1.22}$ | $69.90^{\pm0.38}$ | $65.62^{\pm0.97}$ | $63.64^{\pm3.00}$ |
| Ours w/o PP | $51.62^{\pm0.79}$ | $56.61^{\pm1.31}$ | $45.50^{\pm0.17}$ | $77.13^{\pm0.46}$ | $51.23^{\pm1.01}$ | $68.11^{\pm1.21}$ | $68.70^{\pm1.21}$ | $61.26^{\pm0.85}$ |
| Ours w/o P-SSL | $70.88^{\pm0.21}$ | $71.78^{\pm0.28}$ | $\mathbf{61.28^{\pm0.66}}$ | $86.08^{\pm0.14}$ | $66.79^{\pm0.22}$ | $\mathbf{94.02^{\pm0.27}}$ | $86.31^{\pm0.10}$ | $73.30^{\pm0.10}$ |
| Ours | $\mathbf{71.35^{\pm0.32}}$ | $\mathbf{74.93^{\pm0.16}}$ | $57.48^{\pm1.28}$ | $\mathbf{87.98^{\pm0.91}}$ | $\mathbf{67.72^{\pm0.11}}$ | $90.31^{\pm0.17}$ | $\mathbf{89.97^{\pm0.41}}$ | $\mathbf{75.25^{\pm0.13}}$ |
| Fine-tuning (FT) | $75.02^{\pm0.09}$ | $76.69^{\pm0.40}$ | $\mathbf{65.59^{\pm0.60}}$ | $86.67^{\pm0.39}$ | $70.27^{\pm0.99}$ | $95.76^{\pm0.05}$ | $87.73^{\pm0.05}$ | $75.79^{\pm0.30}$ |
| FT + P-SSL | $\mathbf{75.93^{\pm0.44}}$ | $\mathbf{80.46^{\pm0.16}}$ | $62.48^{\pm0.97}$ | $\mathbf{87.32^{\pm1.15}}$ | $\mathbf{71.00^{\pm0.35}}$ | $\mathbf{96.59^{\pm0.37}}$ | $\mathbf{91.48^{\pm0.32}}$ | $\mathbf{78.53^{\pm0.28}}$ |
| R-SSL with ImageNet | $52.22^{\pm1.12}$ | $55.54^{\pm1.23}$ | $41.84^{\pm3.64}$ | $75.36^{\pm0.80}$ | $52.13^{\pm1.98}$ | $68.55^{\pm1.68}$ | $66.34^{\pm1.07}$ | $59.93^{\pm1.25}$ |
| FT + R-SSL | $76.16^{\pm0.16}$ | $80.38^{\pm0.16}$ | $61.52^{\pm0.40}$ | $87.08^{\pm0.51}$ | $61.52^{\pm0.83}$ | $96.20^{\pm0.04}$ | $90.33^{\pm0.11}$ | $78.27^{\pm0.19}$ |

Table 14: Top-1 accuracy in various target dataset sizes

| | Samping Rate | | |
|---|---|---|---|
| | 25% | 50% | 75% |
| Scratch | $6.14^{\pm0.42}$ | $40.69^{\pm1.50}$ | $67.46^{\pm2.93}$ |
| Logit Matching | $10.39^{\pm1.43}$ | $63.14^{\pm1.51}$ | $79.64^{\pm1.83}$ |
| Soft Target | $4.50^{\pm1.93}$ | $48.71^{\pm7.99}$ | $72.08^{\pm1.27}$ |
| Ours w/o PP | $11.73^{\pm0.88}$ | $45.27^{\pm0.92}$ | $69.21^{\pm0.88}$ |
| Ours w/o P-SSL | $59.48^{\pm1.23}$ | $82.46^{\pm0.36}$ | $88.31^{\pm0.10}$ |
| Ours | $\mathbf{61.90^{\pm0.60}}$ | $\mathbf{82.65^{\pm0.30}}$ | $\mathbf{89.33^{\pm0.19}}$ |

the generative model is quite different from one of the classification tasks and that our method is well suited to absorb such a difference between tasks for knowledge transfer.

### C.1.2 COMPARSION OF RN18 AND CUSTOM ARCHITECTURE

We provide further comparison of RN-18 architecture and the custom architecture found by the architecture search in Sec. 4.2. Here, we tested RN18 with our methods (PP, P-SSL, and PP+P-SSL) on the same setting as Sec. 4.2 to confirm the superiority of the custom architecture. We summarize the results in Table 16. This shows that the custom architecture is superior to the original RN18 $(4, 4, 4, 4)$ in all settings, and also indicates that this is a fair comparison among methods.

### C.2 TARGET DATASET SIZE

We evaluate the performance of our method on a limited target data setting. We randomly sampled the subsets of the training dataset (StanfordCars) for each sampling rate of 25, 50, and 75%, and used them to train target models with our method. Note that the pseudo datasets were created by using the reduced datasets. Table 14 lists the results. We observed that our method can outperform the baselines in all sampling rate settings.

### C.3 SOURCE GENERATIVE MODELS

Table 17 lists the results with the architectures of multiple generative models. We tested our method by using pseudo samples generated from the generative models for multiple resolutions including SNGAN (Miyato et al., 2018), SAGAN (Zhang et al., 2019), and ADM-G (Dhariwal & Nichol, 2021). We implemented these generative models on the basis of open source repositories including `pytorch-pretrained-BigGAN`[3], `Pytorch-StudioGAN`[4] by Minguk et al. (2021), and `guided-diffusion`[5] by Dhariwal & Nichol (2021); we used the pre-trained weights distributed by the repositories. We measured the top-1 test accuracy on the target task (StanfordCars) and Fréchet Inception Distance (FID, Heusel et al. (2017)). In Table 17, the generative models with better FID scores tended to achieve a higher top-1 accuracy score with PP and P-SSL. Regarding

---

[3]https://github.com/huggingface/pytorch-pretrained-BigGAN

[4]https://github.com/POSTECH-CVLab/PyTorch-StudioGAN

[5]https://github.com/openai/guided-diffusion

Table 15: Comparison of RN18 and Custom RN

| Architectures | Top-1 Acc. (%) |
|---|---|
| RN18 $(4, 4, 4, 4)$ | $73.27^{\pm 0.43}$ |
| CRN $(2, 2, 8, 2)$ | $76.10^{\pm 1.10}$ |
| CRN $(2, 2, 10, 2)$ | $75.65^{\pm 1.60}$ |
| CRN $(2, 10, 2, 2)$ | $\mathbf{77.01^{\pm 0.57}}$ |
| CRN $(6, 2, 6, 2)$ | $76.76^{\pm 1.58}$ |
| CRN $(8, 4, 2, 2)$ | $76.78^{\pm 1.72}$ |

Table 16: Comparison of RN18 and Custom RN

| | Top-1 Acc. (%) | |
|---|---|---|
| | RN18 $(4, 4, 4, 4)$ | Custom RN18 $(2, 10, 2, 2)$ |
| Scratch | $73.27^{\pm 0.43}$ | $\mathbf{77.01^{\pm 0.57}}$ |
| Fine-tuning | $87.75^{\pm 0.35}$ | N/A |
| P-SSL | $77.56^{\pm 0.62}$ | $\mathbf{80.86^{\pm 0.16}}$ |
| PP | $86.09^{\pm 0.41}$ | $\mathbf{88.75^{\pm 0.33}}$ |
| PP+P-SSL | $87.61^{\pm 0.41}$ | $\mathbf{89.13^{\pm 0.23}}$ |

Table 17: Comparison of multiple source generative models (StanfordCars)

| | Top-1 Acc. (%) | | |
|---|---|---|---|
| **Source Generative Model** | PP | P-SSL | FID$(\mathcal{D}_{s \leftarrow t}, \mathcal{D}_t)$ |
| **$128 \times 128$ resolution** | | | |
| SNGAN (Miyato et al., 2018) | $87.07^{\pm 0.26}$ | $72.75^{\pm 2.60}$ | 108.31 |
| SAGAN (Zhang et al., 2019) | $89.10^{\pm 0.19}$ | $77.66^{\pm 2.15}$ | 58.67 |
| BigGAN (Brock et al., 2019) | $89.97^{\pm 0.27}$ | $77.54^{\pm 1.45}$ | 27.17 |
| **$256 \times 256$ resolution** | | | |
| BigGAN (Brock et al., 2019) | $90.25^{\pm 0.19}$ | $\mathbf{80.01^{\pm 0.14}}$ | 19.92 |
| ADM-G (Dhariwal & Nichol, 2021) | $90.23^{\pm 1.06}$ | $79.44^{\pm 0.47}$ | 18.72 |
| **$512 \times 512$ resolution** | | | |
| BigGAN (Brock et al., 2019) | $90.14^{\pm 0.24}$ | $78.38^{\pm 1.61}$ | **17.87** |

the resolution, the models of $256 \times 256$, the generated samples of which are the nearest to the input size of $C_t^{\mathcal{A}_t}$ ($224 \times 224$), were the best. From these results, we recommend using generative models synthesizing high-fidelity samples at a resolution close to the target models when applying our method.

## C.4 ANALYSIS OF PSEUDO PRE-TRAINING

We analyze the characteristics of PP by varying the synthesized samples.

### C.4.1 SYNTHESIZING STRATEGY

We compare the four strategies using the source generative models for PP. We tested **Uniform**: synthesizing samples for all source classes (default), **Filtered**: synthesizing samples for target-related source classes identified by the same protocol as in Sec. 4.5, **PCS**: synthesizing samples by PCS, **Offline**: synthesizing fixed samples in advance of training and training $C_s^{\mathcal{A}_t}$ with them instead of sampling from $G_s$. In PCS, we optimized the pre-training models with pseudo source soft labels generated in the process of PCS. Table 18 shows the target task performances. We found that the Uniform model achieved the best performance. We can infer that, in PP, pre-training with various classes and samples is more important than that with only target-related classes or fixed samples.

## C.5 ANALYSIS OF PSEUDO SEMI-SUPERVISED LEARNING

### C.5.1 SEMI-SUPERVISED LEARNING ALGORITHM

We compare SSL algorithms in P-SSL. We used six SSL algorithms: EntMin (Grandvalet & Bengio, 2005), Pseudo Label (Lee et al., 2013), Soft Pseudo Label, Consistency Regularization, FixMatch (Sohn et al., 2020), and UDA (Xie et al., 2020). Soft Pseudo Label is a variant of Pseudo Label, which uses the sharpen soft pseudo labels instead of the one-hot (hard) pseudo labels. Consistency Regularization computes the unsupervised loss of UDA without the confidence thresholding. Table 19 shows the results on StanfordCars, where Pseudo Supervised is a model using a pairs of $(x_{s \leftarrow t}, y_t)$ in pseudo conditional sampling for supervised training. UDA achieved the best result.

Table 18: Comparison of pseudo pre-training variants

|  | Top-1 Acc. (%) |
| --- | --- |
| Uniform | $\mathbf{90.95^{\pm 0.21}}$ |
| Filtered | $84.89^{\pm 0.23}$ |
| PCS | $85.90^{\pm 0.35}$ |
| Offline (1.3M) | $89.08^{\pm 0.20}$ |

Table 19: Comparison of algorithms for P-SSL ($\mathcal{A}_t$: WRN-50-2)

|  | Top-1 Acc. (%) |
| --- | --- |
| Scratch | $76.21^{\pm 1.40}$ |
| Pseudo Supervised | $53.03^{\pm 1.79}$ |
| EntMin (Grandvalet & Bengio, 2005) | $72.56^{\pm 3.33}$ |
| Pseudo Label (Lee et al., 2013) | $74.49^{\pm 2.26}$ |
| Soft Pseudo Label | $78.44^{\pm 2.41}$ |
| Consistency Regularization | $79.17^{\pm 1.79}$ |
| FixMatch (Sohn et al., 2020) | $74.31^{\pm 3.27}$ |
| UDA (Xie et al., 2020) | $\mathbf{80.01^{\pm 0.14}}$ |

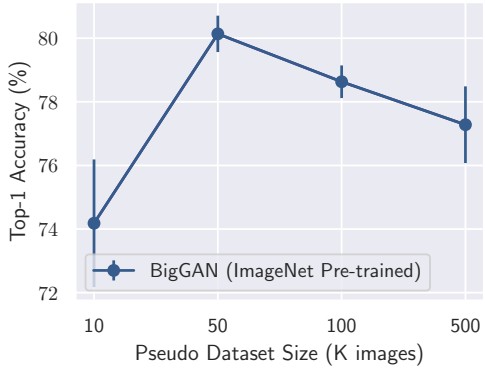

Figure 4: Top-1 accuracy of P-SSL when scaling pseudo dataset size

More importantly, the methods using hard labels (Pseudo Supervised, Pseudo Label, and FixMatch) failed to outperform the scratch models, whereas the soft label based methods improved the performance. This indicates that translating the label of pseudo samples as the interpolation of the target labels can improve the performance as mentioned in Sec. 3.2.3.

### C.5.2 SAMPLE SIZE

We evaluate the effect of the sizes of pseudo datasets for P-SSL on the target test accuracy. We varied the pseudo dataset sizes in {10K, 50K, 100K, 500K } and tested the target performance of P-SSL on the StanfordCars dataset, as shown in Figure 4 (right). We found that the middle range of the dataset size (50K and 100K images) achieved better results. This suggests that P-SSL does not require generating extremely large pseudo datasets for boosting the target models.

### C.5.3 OUTPUT LABEL FUNCTION

We discuss the performance comparison of output label functions in PCS. The output label function is crucial for synthesizing the target-related samples from source generative models since it directly determines the attributes on the pseudo samples. We tested six labeling strategies, i.e., **Random Label**: attaching uniformly sampled source labels, **Softmax**: using softmax outputs of $C_s^{\mathcal{A}_s}$ (default), **Temperature Softmax**: applying temperature scaling to output logits of $C_s^{\mathcal{A}_s}$ and using the softmax output, **Argmax**: using one-hot labels generated by selecting the class with the maximum probability in the softmax output of $C_s^{\mathcal{A}_s}$, **Sparsemax** (Martins & Astudillo, 2016): computing the Euclidean projections of the logit of $C_t$ representing sparse distributions in the source label space, and **Classwise Mean**: computing the mean of softmax outputs of $C_s^{\mathcal{A}_s}$ for each target class and using it as representative pseudo source labels of the target class to generate pseudo samples. Table 20 shows the comparison of the labeling strategies. Among the strategies, Softmax is the best choice for PCS in terms of the target performance (top-1 accuracy) and the relatedness toward target datasets (FID). This means that the pseudo source label $y_{s \leftarrow t}$ by Softmax succeeds in representing the characteristics of a target sample $x_t$ and its form of the soft label is important to extract target-related information via a source generative model $G_s$.

Table 20: Comparison of output label functions in PCS

|  | Top-1 Acc. (%) | FID |
|---|---|---|
| Random Label | $75.55^{\pm 0.35}$ | 134.37 |
| Softmax | $\mathbf{80.01^{\pm 0.14}}$ | **19.92** |
| Temperature Softmax ($\tau = 0.4$) | $79.94^{\pm 1.12}$ | 20.68 |
| Argmax | $78.89^{\pm 2.01}$ | 22.35 |
| Sparsemax | $76.08^{\pm 1.14}$ | 24.28 |
| Classwise Mean | $78.56^{\pm 2.83}$ | 22.14 |

Table 21: Effect of $\tau$ in UDA

| $\tau$ | Top-1 Acc. (%) |
|---|---|
| 0.2 | $79.59^{\pm 1.59}$ |
| 0.4 | $80.01^{\pm 0.14}$ |
| 0.6 | $80.43^{\pm 0.25}$ |
| 0.8 | $\mathbf{81.81^{\pm 0.58}}$ |
| 1.0 | $79.16^{\pm 0.25}$ |

### C.5.4 SEARCHING TEMPERATURE HYPERPARAMETER FOR UDA

Here, we investigate the effect of hyperparameter $\tau$ for UDA, which is mainly used in the main paper. By the definition of temperature softmax function $(\exp(y_i/\tau)/\sum_j^K \exp(y_j/\tau))$, the temperature parameter $\tau$ controls the sharpness of the predicted class-conditional distribution; lower temperature outputs sharper distribution. In this regard, in our P-SSL, $\hat{C}_t(x_{s \leftarrow t}, \tau; \theta)$, which determines how to represent the pseudo sample $x_{s \leftarrow t}$ with the soft target class labels, can be changed by $\tau$. Thus, the choice of $\tau$ can be an important factor for training in P-SSL. To evaluate the effect, we tested P-SSL by varying $\tau$ on StanfordCars as shown in Table 21 The results show that the moderately higher $\tau$ achieved better target performances. This suggests that representing $x_{s \leftarrow t}$ by softer target class labels can bring a positive effect to target classifiers, which is consistent with the discussions in Sec. 3.2.1 and 3.2.3.

### C.5.5 P-SSL VS. SSL USING TARGET LABELED DATA

We provide further analysis to confirm the performance of the unlabeled pseudo samples $x_{s \leftarrow t}$ by comparing with a SSL method using real target samples in the unsupervised loss function. We call this method as T-SSL, and T-SSL can be another simple baseline for P-SSL because it can simply discard pseudo sample synthesis in P-SSL. The results are shown in Table 22. We can see that applying supervised loss and unsupervised loss to the same sample has a negative effect: the use of pseudo-labels to target labeled data might

Table 22: P-SSL vs. T-SSL

|  | Top-1 Acc. (%) |
|---|---|
| PP | $90.95^{\pm 0.21}$ |
| PP+T-SSL | $87.65^{\pm 0.35}$ |
| PP+P-SSL | $\mathbf{91.76^{\pm 0.41}}$ |

promote overfitting. This result indicates that $x_{s \leftarrow t}$ in P-SSL certainly performs as useful unlabeled data in UDA than directly using target data.

## D QUALITATIVE EVALUATION OF PSEUDO SAMPLES

We discuss qualitative studies of the pseudo samples generated by PCS. To confirm the correspondences between the target and pseudo samples, we used StanfordCars as the target dataset and generated samples from BigGAN with the same setting as in Sec. 4. Figure 5 shows the visualizations of the source dataset (ImageNet), target dataset (StanfordCars), and pseudo samples generated by PCS. The samples were randomly selected from each dataset. We can see that PCS succeeded in generating target-related samples from the target samples. To assess the validity of using pseudo source soft labels in PCS, we analyzed the pseudo samples corresponding to each target label. Figure 6 shows the pseudo samples generated by the target samples of `Hummer` and `Aston Martin V8 Convertible` classes in StanfordCars. We confirm that the pseudo samples by PCS can capture the features of target classes. This also can confirm the ranking of the confidence scores for source classes listed in Table 23; the pseudo source soft labels seem to represent the target samples by the interpolation of source classes.

ImageNet (target-related)    StanfordCars    PCS

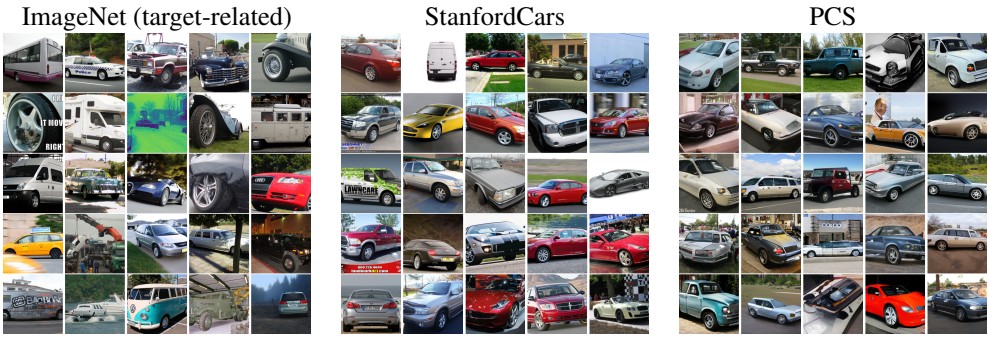

Figure 5: Samples of source, target, and pseudo datasets (random picking).

StanfordCars    PCS

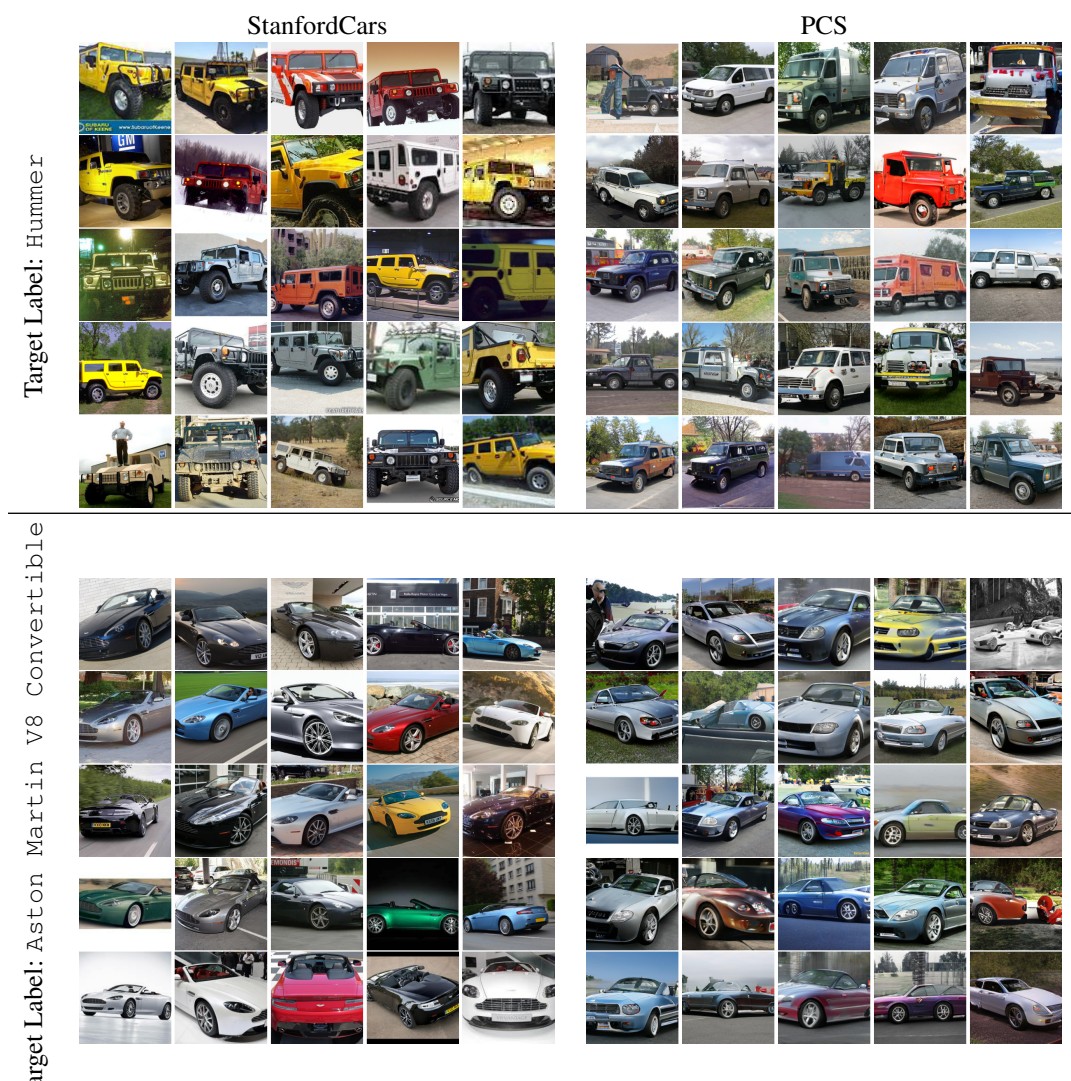

Figure 6: Pseudo samples generated by PCS (random picking)

Table 23: Ranking of averaged confidence scores of ImageNet classes corresponding to target classes

| | Target Class Label | | |
|---|---|---|---|
| Rank | All target classes | Hummer | Aston Martin V8 Convertible |
| 1st | sports car (23.5%) | jeep (41.0%) | convertible (62.8%) |
| 2nd | beach wagon (15.8%) | limousine (11.1%) | sports car (27.4%) |
| 3rd | minivan (12.0%) | snowplow (6.7%) | racer (1.3%) |
| 4th | convertible (10.0%) | moving van (6.7%) | pickup truck (1.3%) |
| 5th | pickup truck (7.0%) | tow truck (6.3%) | car wheel (1.3%) |

Table 24: Corresponding ImageNet classes to target datasets (1)

| Target Dataset | ImageNet Classes |
|---|---|
| Caltech-256-60 | goldfish, electric ray, ostrich, great grey owl, tree frog, tailed frog, loggerhead, leatherback turtle, common iguana, triceratops, trilobite, scorpion, barn spider, tick, centipede, hummingbird, drake, goose, tusker, wallaby, jellyfish, nematode, conch, snail, rock crab, fiddler crab, American lobster, isopod, black stork, American egret, king penguin, killer whale, whippet, Saluki, leopard, jaguar, cheetah, brown bear, American black bear, fly, grasshopper, cockroach, mantis, monarch, starfish, sea urchin, porcupine, sorrel, zebra, ibex, hartebeest, Arabian camel, llama, skunk, gorilla, chimpanzee, Indian elephant, African elephant, acoustic guitar, airliner, airship, altar, analog clock, assault rifle, backpack, balloon, ballpoint, Band Aid, barbell, barrow, bathtub, beacon, beaker, bearskin, beer glass, bell cote, bicycle-built-for-two, binder, binoculars, bolo tie, bottlecap, bow, brass, breastplate, buckle, candle, cannon, canoe, can opener, carpenter's kit, car wheel, cassette, cassette player, CD player, chain, chest, chime, clog, coffee mug, coffeepot, coil, combination lock, corkscrew, cowboy hat, cradle, crane, crash helmet, croquet ball, desktop computer, dial telephone, doormat, drilling platform, drum, dumbbell, electric fan, electric guitar, envelope, face powder, fire engine, fire screen, flagpole, folding chair, football helmet, fountain, French horn, frying pan, gasmask, gas pump, goblet, golf ball, gong, grand piano, guillotine, hair slide, hamper, hand blower, hand-held computer, handkerchief, harmonica, harp, harvester, hook, horse cart, hourglass, iPod, jersey, jigsaw puzzle, joystick, knee pad, ladle, lampshade, laptop, lawn mower, letter opener, lighter, loudspeaker, loupe, lumbermill, magnetic compass, mailbag, mailbox, maraca, marimba, maze, microphone, microwave, missile, modem, moped, mortar, mosque, mountain bike, mountain tent, mouse, mousetrap, muzzle, nail, neck brace, nipple, notebook, obelisk, ocarina, oil filter, oscilloscope, oxygen mask, packet, paddle, palace, parachute, park bench, pay-phone, pedestal, pencil sharpener, perfume, Petri dish, photocopier, pick, pier, pill bottle, ping-pong ball, pitcher, plane, pole, pool table, pot, printer, projectile, projector, puck, punching bag, quill, racket, radio, radio telescope, reel, refrigerator, revolver, rifle, rubber eraser, rule, running shoe, safety pin, sandal, scabbard, scale, school bus, schooner, screen, screwdriver, shield, ski, slot, snowmobile, soap dispenser, soccer ball, sock, solar dish, sombrero, space bar, spatula, speedboat, spotlight, stethoscope, stopwatch, stretcher, studio couch, sunglass, sunscreen, suspension bridge, swing, switch, syringe, table lamp, tape player, teapot, teddy, tennis ball, thimble, thresher, toaster, tobacco shop, toilet seat, torch, tow truck, tray, tricycle, tripod, tub, typewriter keyboard, umbrella, unicycle, upright, vase, waffle iron, wall clock, wallet, warplane, washer, water jug, whiskey jug, whistle, Windsor tie, wine bottle, wool, worm fence, yawl, web site, comic book, street sign, traffic light, book jacket, plate, cheeseburger, hotdog, spaghetti squash, fig, carbonara, red wine, cup, eggnog, cliff, geyser, lakeside, promontory, seashore, valley, volcano, daisy, hip, earthstar, hen-of-the-woods |
| CUB-200-2011 | hen, brambling, goldfinch, house finch, junco, indigo bunting, robin, bulbul, jay, magpie, chickadee, water ouzel, kite, bald eagle, vulture, great grey owl, black grouse, ptarmigan, ruffed grouse, prairie chicken, quail, partridge, macaw, lorikeet, coucal, bee eater, hornbill, hummingbird, jacamar, toucan, drake, red-breasted merganser, goose, black swan, white stork, black stork, spoonbill, little blue heron, American egret, bittern, crane, limpkin, European gallinule, American coot, bustard, ruddy turnstone, red-backed sandpiper, redshank, dowitcher, oystercatcher, pelican, king penguin, albatross, worm fence |
| DTD | electric ray, stingray, leatherback turtle, thunder snake, hognose snake, horned viper, sidewinder, trilobite, harvestman, barn spider black widow tick, jellyfish, sea anemone, brain coral, flatworm, nematode, conch, sea slug, chiton, chambered nautilus, fiddler crab, isopod, komondor, tiger, leaf beetle, dung beetle, bee, ant, walking stick, cockroach, sea urchin, sea cucumber, zebra, apron, backpack, bakery, balloon, Band Aid, bannister, bath towel, beer glass, bib, binder, bonnet, bottlecap, bow tie, breastplate, broom, buckle, candle, cardigan, chain, chainlink fence, chain mail, cliff dwelling, cloak, coil, confectionery, crate, cuirass, dishrag, dome, doormat, envelope, face powder, feather boa, fire screen, fountain, fur coat, golf ball, gown, hair slide, hamper, handkerchief, honeycomb, hook, hoopskirt, jean, jersey, jigsaw puzzle, knot, lampshade, lighter, loudspeaker, mailbag, manhole cover, mask, matchstick, maze, megalith, microphone, mitten, mosquito net, nail, necklace, overskirt, packet, padlock, paintbrush, pajama, paper towel, pencil box, Petri dish, picket fence, pillow, pinwheel, plastic bag, poncho, pot, prayer rug, prison, purse, quill, quilt, radiator, radio, rubber eraser, rule, safety pin, saltshaker, sarong, screw, shield, shoji, shopping basket, shovel, shower cap, shower curtain, sleeping bag, solar dish, space heater, spider web, stole, stone wall, strainer, swab, sweatshirt, swimming trunks, switch, syringe, tennis ball, thatch, theater curtain, thimble, tile roof, tray, trench coat, umbrella, vase, vault, velvet, waffle iron, wall clock, wallet, wardrobe, water bottle, wig, window screen, window shade, Windsor tie, wooden spoon, wool, web site, crossword puzzle, book jacket, trifle, ice cream, ice lolly, French loaf, pretzel, head cabbage, broccoli, cauliflower, strawberry, lemon, fig, jackfruit, custard apple, pomegranate, hay, chocolate sauce, dough, meat loaf, potpie, cup, eggnog, bubble, cliff, coral reef, geyser, sandbar, valley, volcano, corn, buckeye, coral fungus, hen-of-the-woods, ear, toilet tissue |
| FGVC-Aircraft | aircraft carrier, airliner, airship, missile, projectile, space shuttle, speedboat, trimaran, warplane, wing |
| Indoor67 | academic gown, altar, bakery, balance beam, bannister, barbell, barber chair, barbershop, barrel, bathing cap, bathtub, beer bottle, bookcase, bookshop, bullet train, butcher shop, carousel, carton, cash machine, china cabinet, church, cinema, coil, confectionery, cradle, crate, crib, crutch, desk, desktop computer, dining table, dishwasher, dome, drum, dumbbell, electric locomotive, entertainment center, file, fire screen, folding chair, forklift, fountain, four-poster, golfcart, grand piano, greenhouse, grocery store, guillotine, home theater, horizontal bar, jigsaw puzzle, lab coat, laptop, library, limousine, loudspeaker, lumbermill, marimba, maze, medicine chest, microwave, minibus, monastery, monitor, mosquito net, organ, oxygen mask, palace, parallel bars, passenger car, patio, photocopier, pier, ping-pong ball, planetarium, plate rack, pole, pool table, pot, potter's wheel, prayer rug, printer, prison, projector, quilt, radio, refrigerator, restaurant, rocking chair, rotisserie, scoreboard, screen, shoe shop, shoji, shopping basket, shower curtain, sliding door, slot, solar dish, spotlight, stage, steel arch bridge, stove, streetcar, stretcher, studio couch, table lamp, tape player, television, theater curtain, throne, tobacco shop, toilet seat, toyshop, tripod, tub, turnstile, upright, vacuum, vault, vending machine, vestment, wardrobe, washbasin, washer, window screen, window shade, wine bottle, wok, comic book, plate, groom |

Table 25: Corresponding ImageNet classes to target datasets (2)

| Target Dataset | ImageNet Classes |
| --- | --- |
| OxfordFlower | goldfish, cock, harvestman, garden spider, hummingbird, sea anemone, conch, snail, slug, sea slug, chambered nautilus, ladybug, long-horned beetle, leaf beetle, fly, bee, ant, grasshopper, cricket, walking stick, mantis, lacewing, admiral, ringlet, monarch, cabbage butterfly, sulphur butterfly, lycaenid, sea urchin, birdhouse, bonnet, candle, chainlink fence, greenhouse, hair slide, hamper, handkerchief, lotion, paper towel, perfume, picket fence, pinwheel, plastic bag, pot, quill, shower cap, tray, umbrella, vase, velvet, wool, head cabbage, cauliflower, zucchini, spaghetti squash, acorn squash, butternut squash, cucumber, artichoke, bell pepper, cardoon, mushroom, strawberry, orange, lemon, fig, pineapple, custard apple, pomegranate, coral reef, rapeseed, daisy, yellow lady's slipper, corn, acorn, hip, buckeye, coral fungus, stinkhorn, earthstar, hen-of-the-woods, ear |
| OxfordPets | Chihuahua, Japanese spaniel, Maltese dog, Pekinese, Shih-Tzu, Blenheim spaniel, papillon, toy terrier, Rhodesian ridgeback, basset, beagle, bloodhound, bluetick, Walker hound, English foxhound, redbone, Italian greyhound, Ibizan hound, Norwegian elkhound, Weimaraner, Staffordshire bullterrier, American Staffordshire terrier, Irish terrier, Norwich terrier, Yorkshire terrier, Lakeland terrier, cairn, Australian terrier, Dandie Dinmont, Boston bull, Scotch terrier, Tibetan terrier, silky terrier, soft-coated wheaten terrier, West Highland white terrier, Lhasa, flat-coated retriever, golden retriever, Labrador retriever, Chesapeake Bay retriever, German short-haired pointer, English setter, Brittany spaniel, English springer, Welsh springer spaniel, cocker spaniel, kuvasz, schipperke, groenendael, kelpie, German shepherd, miniature pinscher, boxer, bull mastiff, Tibetan mastiff, French bulldog, Great Dane, Saint Bernard, Eskimo dog, Siberian husky, affenpinscher, basenji, pug, Leonberg, Newfoundland, Great Pyrenees, Samoyed, Pomeranian, chow, keeshond, Brabancon griffon, Pembroke, toy poodle, miniature poodle, Mexican hairless, white wolf, dingo, tabby, tiger cat, Persian cat, Siamese cat, Egyptian cat, lynx, bath towel, bucket, carton, plastic bag, quilt, sleeping bag, space heater, tennis ball, window screen |
| StanfordCars | ambulance, amphibian, beach wagon, cab, car wheel, convertible, grille, jeep, minibus, limousine, minivan, mobile home, moving van, parking meter, passenger car, pickup, police van, racer, recreational vehicle, snowplow, sports car, tow track, trailer truck |
| StanfordDogs | Chihuahua, Japanese spaniel, Maltese dog, Pekinese, Shih-Tzu, Blenheim spaniel, papillon, toy terrier, Rhodesian ridgeback, Afghan hound, basset, beagle, bloodhound, bluetick, black-and-tan coonhound, Walker hound, English foxhound, redbone, borzoi, Irish wolfhound, Italian greyhound, whippet, Ibizan hound, Norwegian elkhound,otterhound, Saluki, Scottish deerhound, Weimaraner, Staffordshire bullterrier, American Staffordshire terrier, Bedlington terrier, Border terrier, Kerry blue terrier, Irish terrier, Norfolk terrier,Norwich terrier, Yorkshire terrier, wire-haired fox terrier, Lakeland terrier, Sealyham terrier, Airedale, cairn, Australian terrier, Dandie Dinmont, Boston bull, miniature schnauzer, giant schnauzer,standard schnauzer, Scotch terrier, Tibetan terrier, silky terrier, soft-coated wheaten terrier, West Highland white terrier, Lhasa, flat-coated retriever, curly-coated retriever, golden retriever, Labrador retriever,Chesapeake Bay retriever, German short-haired pointer, vizsla, English setter, Irish setter, Gordon setter, Brittany spaniel, clumber, English springer, Welsh springer spaniel, cocker spaniel, Sussex spaniel,Irish water spaniel, kuvasz, schipperke, groenendael, malinois, briard, kelpie, komondor, Old English sheepdog, Shetland sheepdog, collie, Border collie, Bouvier des Flandres, Rottweiler, German shepherd,Doberman, miniature pinscher, Greater Swiss Mountain dog, Bernese mountain dog, Appenzeller, EntleBucher, boxer, bull mastiff, Tibetan mastiff, French bulldog, Great Dane, Saint Bernard, Eskimo dog,malamute, Siberian husky, affenpinscher, basenji, pug, Leonberg, Newfoundland, Great Pyrenees, Samoyed, Pomeranian, chow, keeshond, Brabancon griffon, Pembroke, Cardigan, toy poodle, miniature poodle, standard poodle, Mexican hairless, dingo, dhole, African hunting dog |

