# OpenReview forum: "Transfer Learning with Pre-trained Conditional Generative Models"
_ICLR.cc/2023/Conference — Submitted to ICLR 2023_

### Official Review · Reviewer_kCqh · 2022-10-25

**Confidence:** 3
**Correctness:** 3
**Technical Novelty And Significance:** 2
**Empirical Novelty And Significance:** 2
**Recommendation:** 5

**Clarity, Quality, Novelty And Reproducibility:**

The overall writing is clear and easy to follow.

The only thing confused me is Eq (3). In Eq (3), $y_{s\leftarrow t}$ is generated from $C_{s}^{A_t}$.
However, in Line 4 of Algorithm 1, $y_{s\leftarrow t}$ is generated from $\hat{C}_{s}^{A_t}$.

Additionally, in the last paragraph of Page 4, the last layer of $C_{s}^{A_s}$ is swapped.
But in Line 2 of Algorithm 1, the last layer of $C_{s}^{A_t}$ is swapped.

This work didn't provide source code but I believe it should be reproducible.

**Strength And Weaknesses:**

Strength:
1. The proposed transfer learning in the three assumptions is difficult to learn but the authors make it work.
2. The author found directly using the original label for the pseudo data in P-SSL didn't work and UDA as the unsupervised loss can achieve better performance. This observation is important and may help other related research works.

Weaknesses:
1. I'm afraid the main contribution is UDA. Because UDA loss is used, the author can directly apply UDA loss on the original target data $x_t$ rather than $x_{s\leftarrow t}$. In this way, the pseudo data construction is basically discarded. Although the authors indicate there is less prior work in this area and compare with two simple baselines, I think this is also an important and simple baseline.
2. I think a more rigorous experiment setup is needed or should be added. For example, in sec 4.6, both the source and target dataset are vehicle or cars related. According the proposed transfer learning with three assumptions, the source data can be vehicle related, and the target dataset could be animal related.

**Summary Of The Paper:**

This paper discussed the transfer learning in relatively relaxed conditions. Particularly, the paper assumes (1) no label overlapping space between source and target tasks, (2) no source dataset access when training the target task, and (3) the inconsistent model architectures.

To tackle the transfer learning problem in above assumptions, the authors propose the pseudo pre-training (PP) and pseudo semi-supervised learning (P-SSL).

During PP stage, the target classifier is pre-trained on the pseudo data generation from source generative model but the final classification layer will be replaced later because the output is in source label space.

During P-SSL, the target data will be first fed into pre-trained target classifier and output a pseudo soft label in source space. The pseudo soft label will be fed into source generative model to produce a pseudo data. This pseudo data will be used for unsupervised training. For unsupervised training, the paper adopted UDA loss rather than the cross entropy loss based on the original label.

The experiments compare the proposed method with scratch training and knowledge distillation.

**Summary Of The Review:**

In general, the paper raises an important and difficult topic of transfer learning, and proposes a pseudo data based framework to tackle this problem. However, the experiments can not fully verify the proposed method. My initial review is marginally below the acceptance threshold.

---

> ### Author Response · Authors · 2022-11-11
> **First response to Reviewer kCqh**
>
> We sincerely appreciate your elaborate reading of our paper and your detailed insightful comments.
>
> ## Replies to Weakness 1
> > I'm afraid the main contribution is UDA. Because UDA loss is used, the author can directly apply UDA loss on the original target data $x_t$ rather than $x_{s\leftarrow t}$. In this way, the pseudo data construction is basically discarded. Although the authors indicate there is less prior work in this area and compare with two simple baselines, I think this is also an important and simple baseline.
>
> Thank you for the constructive suggestion! We totally agree with you; applying the target dataset for unsupervised loss of UDA is an important baseline. Since UDA unsupervised loss is usually applied to unsupervised data, we missed the pattern of applying UDA to supervised target data. If we apply UDA to target labeled data (we call this method as T-SSL), we obtain the following results:
>
> Method | Mean test acc. $\pm$ std.
> -- | --
> PP | 90.95 $\pm$ 0.21
> PP+T-SSL | 87.65 $\pm$ 0.35
> PP+P-SSL | 91.76 $\pm$ 0.41
>
> We can see that applying UDA loss to the (originally labeled) target data has a negative effect: the use of pseudo-labels to target labeled data might promote overfitting. This result indicates that $x_{s\leftarrow t}$ in P-SSL certainly performs as useful unlabeled data in UDA than directly using target data. Thus, the main contribution of the accuracy gain is not applying UDA itself. We have added this discussion and results in Appendix C.5.5.
>
> ## Replies to Weakness 2
> > I think a more rigorous experiment setup is needed or should be added. For example, in sec 4.6, both the source and target dataset are vehicle or cars related. According the proposed transfer learning with three assumptions, the source data can be vehicle related, and the target dataset could be animal related.
>
> Thank you for the suggestion of a valid experiment. In Sec. 4.6, we have chosen the generative model of CompCars as $G_s$ because it can be considered to satisfy Assumption 3.1, which says the pseudo samples $x_{s\leftarrow t}$ should approximate the target distribution. Even so, validating the cases that $G_s$ cannot satisfy Assumption 3.1 is useful. To evaluate such cases, we tested the case that the source is CompCars and the target is OxfordPets. The results are shown in the following table, which clearly shows that knowledge transfer is difficult when there is less relationship between the target and source. This could be due to Assumption 3.1 is not satisfied well. In fact, the FID is also much larger than the case where ImageNet is the source. Therefore, to maximize the gain from the proposed method, we should choose the off-the-shelf generative model $G_s$ pre-trained on general or target-related source datasets.
>
> Method | $\mathcal{D}_s$ :ImageNet | $\mathcal{D}_s$ :CompCars
> -- | -- | --
> PP | 86.31 $\pm$ 0.10 | 74.02 $\pm$ 0.72
> P-SSL | 68.70 $\pm$ 1.21 | 57.44 $\pm$ 4.43
> PP+P-SSL | 89.97 $\pm$ 0.41 | 57.69 $\pm$ 0.85
>
> Metric | $\mathcal{D}_s$ :ImageNet | $\mathcal{D}_s$ :CompCars
> -- | -- | --
> $\text{FID}(\mathcal{D}_{s\leftarrow t}, \mathcal{D}_t)$ | 16.94 | 170.70
>
> ## Replies to other comments
> > The only thing confused me is Eq (3). In Eq (3), $y_{s\leftarrow t}$ is generated from
> $C_s^{A_t}$. However, in Line 4 of Algorithm 1, $y_{s\leftarrow t}$ is generated from $\hat{C_s^{A_t}}$.
> > Additionally, in the last paragraph of Page 4, the last layer of $C_s^{A_s}$ is swapped. But in Line 2 of Algorithm 1, the last layer of $C_s^{A_s}$ is swapped.
>
> Thank you for your comments and we deeply apologize for the confusion due to misrepresentation. First of all, P-SSL uses $C_s^{A_s}$ to generate $y_{s\leftarrow t}$. Thus, the notations of Eq. (3), Eq. (4), and algorithm 1 are all incorrect. We have corrected the notations in the revised papers.
>
> > This work didn't provide source code but I believe it should be reproducible.
>
> Sorry for the inconvenience. We have already provided the code for reviewers privately (please see the above general comment). We plan to upload the code on GitHub when the paper gets accepted.

---

> ### Author Response · Authors · 2022-11-17
> **Gently reminder for phase 1 discussion to Reviewer kCqh (ends Nov. 18)**
>
> Thank you again for your constructive suggestions for experiments! We sincerely remind you that phase 1 of the discussion period ends on Nov. 18 (Next Friday). Have our additional experiments been able to address your concerns? If you have any other comments that require additional and larger experiments, we would be very happy to hear from you.

---

### Official Review · Reviewer_MbsB · 2022-10-28

**Confidence:** 4
**Correctness:** 3
**Technical Novelty And Significance:** 4
**Empirical Novelty And Significance:** 3
**Recommendation:** 6

**Clarity, Quality, Novelty And Reproducibility:**

Please check the above section about the discussion on clarity, quality and novelty.

Not sure about the reproducibility. Some details of the experiments are missed.
Are the authors going to release the code when the paper get accepted?

**Strength And Weaknesses:**

Strength:
Based on my knowledge, the authors are the first to apply the pre-trained GAN to transfer learning. It's a novel. The authors carefully design two scenarios with and without source data. In the meantime, the authors conduct extensive experiments on classification and detection.

Writing:

The introduction and the description of the main idea are clear, however, the symbols are a little misleading. Though the reviewer can guess the general meaning, it will be better to make it clear.
1. In equation (2) do supervised loss and unsupervised loss share similar weight parameters?  Where is $\theta$ on the right side of equation (5)?
2. Does \hat(C_t) and C_t share similar parameters? How does $\tau$ select? Does $C_t$ also have a temperature?


Figure:
- Why does C_s^{A_t} has different color in Figure 1(a)(b)? It will be better to describe the color meaning in caption part.
- Is the complete PP algorithm used in experiments is the combination of Figure 1(a) and Figure 2? My understanding is that we can train $A_s$ in Figure 1(a) and fix it in Figure 2 for transfer learning. If my understanding is correct, it will be better to merge Figure 1/2 and use dotted lines (or anything else) to differentiate different training stages.


Questions:
1. Method
The architecture inconsistency part in Table 1 is interesting, but I have some questions about the second point: "no source dataset access".
Did you fine-tune GAN using the source dataset? I understand the case that the source dataset is available. However, when the source dataset is not available and GAN never sees a similar source dataset before, does the algorithm still work? If not, the description of "no source dataset access" is inaccurate, under which case, we can hardly match the space of the source label and the label space (condition) of GAN, not to mention generate augmentations for transfer learning.

2. Experiments
- How to select the custom architecture in Table 4? Does each method select its own architecture independently using the validation set or are they selected by "ours"? What's the meaning of "from (2,2,2,2) to (2,2,2,10)". What's the performance with Architecture RN18(4,4,4,4)?
- It seems that PP contributes the most in Table 6  but harms the performance in Table 9, any comments on this?



**Summary Of The Paper:**

The authors utilize the pre-trained conditional generation model as a data augmentation technique to facilitate transfer learning. Extensive experience done prove the effectiveness of the proposed method.

**Summary Of The Review:**

The idea is novel. But I do think some expression and explanation could be improved.

---

> ### Author Response · Authors · 2022-11-11
> **First response to Reviewer MbsB (1/2)**
>
> We thank you for precisely reading our paper and for your many helpful comments.
>
> ## Replies to comments for Writing
>
> > the symbols are a little misleading
>
> We are sorry for the unclearness of our writing. We have revised the paper according to your comments.
>
> > 1. In equation (2) do supervised loss and unsupervised loss share similar weight parameters?
>
> Yes. We apologize for causing the confusion. As defined in Sec. 2, $\theta$ is a parameter of a target neural network model $f_{\theta}^{\mathcal{A}_t}=C^{\mathcal{A}_t}_t$, and thus we exactly share the parameter $\theta$ between the supervised and unsupervised loss in Eq. (2).
>
> > Where is $\theta$ on the right side of equation (5)?
>
> By the definition in Sec.2 ($f_{\theta}^{\mathcal{A}_t}=C^{\mathcal{A}_t}_t$), $C^{\mathcal{A}_t}_t$ includes the parameter $\theta$, and we omitted $\theta$ in Eq. (5) for simplicity. However, after reading your comments, we noticed that the notation of Eq. (5) was confusing. Therefore, we have re-written Eq. (5) by using $\theta$ in the revision as follows:
>
> $$
> \mathcal{L}_{\text{unsup}}(x,\theta) =1(\max _{y^{\prime}\in\hat{C}^{\mathcal{A}_t}_t(x,\tau;\theta)}y^{\prime}>\beta)\text{CE}(\hat{C}^{\mathcal{A}_t}_t(x,\tau; \theta),C^{\mathcal{A}_t}_t(T(x);\theta)),
> $$
>
> where $C(\cdot;\theta)$ means that $C$ is parameterized by $\theta$.
>
> > Does $\hat{C_t}$ and $C_t$ share similar parameters? .. Does $C_t$ also have a temperature?
>
> The former is yes and the latter is no. $\hat{C_t}$ and $C_t$ differ only in their output functions i.e., $\hat{C}$ has a temperature softmax function with hyperparameter $\tau$ and $C$ has a standard softmax function without any hyperparameters. Thus, $\hat{C_t}$ and $C_t$ share the parameter $\theta$. This is the same as in the original UDA paper.
>
> > How does $\tau$ select?
>
> We fixed $\tau=0.4$, which is the same as the original UDA paper i.e., we did not search $\tau$ at the submission period for keeping reproducibility. By the way, this question provided us with an interesting discovery. By the definition of temperature softmax function ($\exp (y_i/\tau)/\sum^K_j \exp(y_j/\tau)$), the temperature parameter $\tau$ controls the sharpness of the predicted class-conditional distribution; lower temperature outputs sharper distribution. In this regard, in our P-SSL, $\hat{C_t}(x_{s\leftarrow t},\tau;\theta)$, which determines how to represent the pseudo sample $x_{s\leftarrow t}$ with the soft target class labels, can be changed by $\tau$. Thus, the choice of $\tau$ can be an important factor for training in P-SSL. To evaluate the effect, we tested P-SSL by varying $\tau$ on StanfordCars as follows:
>
> $\tau$ | mean test acc. $\pm$ std.
> -- | --
> 0.2 | 79.59 $\pm$ 1.59
> 0.4 | 80.01 $\pm$ 0.14
> 0.6 | 80.43 $\pm$ 0.25
> **0.8** | **81.81 $\pm$ 0.58**
> 1.0 | 79.16 $\pm$ 0.25
>
> The results show that the moderately higher $\tau$ achieved better target performances. This suggests that representing $x_{s\leftarrow t}$ by softer target class labels can bring a positive effect to target classifiers, which is consistent with the discussions in Sec. 3.2.3 and C.5.1 i.e., treating $x_{s\leftarrow t}$ with softer pseudo target labels bring the improvements. We appreciate you for giving us tips to enhance our claim further. We have added this discussion to Appendix C.5.4.
>
> ## Replies to comments for Figures 1 and 2
>
> > Why does $C_s^{A_t}$ has different color in Figure 1(a)(b)? It will be better to describe the color meaning in caption part.
>
> Sorry for confusing you. The color meanings are that: red represents given source models, light blue represents given target models and datasets, and dark blue represents the output of the proposed methods (PP and P-SSL). In the revision, we have modified the colors in Fig. 1 according to the rule, and added these descriptions to the caption.
>
> > Is the complete PP algorithm used in experiments is the combination of Figure 1(a) and Figure 2?
>
> Not exactly. Single PP uses the idea in Figure 1(a) only, and the complete method (PP+P-SSL) uses the ideas of Fig. 1(a) and 1(b) serially. Fig. 2 illustrates *pseudo conditional sampling* for synthesizing $\mathcal{D}_{s\leftarrow t}$ in Fig. 1(b), and thus, to be more precise, PP+P-SSL is a combination of Fig 1(a), 1(b), and 2.
>
> > My understanding is that we can train $\mathcal{A}_s$ in Figure 1(a) and fix it in Figure 2 for transfer learning.
>
> In PP of Fig. 1(a), we train $\mathcal{A}_t$ to build fundamental representations for the target architecture, not $\mathcal{A}_s$. And, in Fig. 2, we use off-the-shelf pre-trained source classifier $C^{\mathcal{A}_s}_s$ with the source architecture $\mathcal{A}_s$. Thus, Fig. 1(a) and 2 are essentially independent. We recognize that this is also confusing. Our apologies.

---

> ### Author Response · Authors · 2022-11-11
> **First response to Reviewer MbsB (2/2)**
>
> ## Replies to Question 1 (Method)
>
> > Did you fine-tune GAN using the source dataset?
>
> No. In this problem setting, as defined in Sec.2, we assume that the pre-trained models are given i.e., they are off-the-shelf. For example, we mainly used ImageNet pre-trained BigGAN, which is provided in huggingface.co. Thus, the proposed method does not include the training process of source pre-trained generative models. We have modified the descriptions of this setting in Sec. 2 to use the term "off-the-shelf".
>
> > However, when the source dataset is not available and GAN never sees a similar source dataset before, does the algorithm still work?
>
> Yes, our method can stably work well without accessing the source dataset. [As mentioned in Strength section by Reviewer JCMy](https://openreview.net/forum?id=5-3YJbVPp6m&noteId=fT-CNC6pm4L), we revealed that PP can stably improve on multiple target datasets (Table 6) when the source dataset is a general dataset (ImageNet). For P-SSL, in Sec. 3.2.4 and 4.4, we investigate how the pseudo sample $x_{s\leftarrow t}$ are related to the target datasets by measuring distribution gaps (FID) between the target and pseudo samples. By the results in Table 2 and Figure 3, we can say that the final target performance when using PP+P-SSL is predictable by assessing the distribution gaps without accessing the source dataset.
>
> ## Replies to Question 2 (Experiments)
>
> > How to select the custom architecture in Table 4?
>
> > Does each method select its own architecture independently using the validation set or are they selected by "ours"?
>
> > What's the meaning of "from (2,2,2,2) to (2,2,2,10)"
>
> Sorry for the unclearness. We greedily selected the best architecture on the scratch training setting for a fair comparison. Thus, the answer to the second question is NO; we did not optimize the custom architecture toward our method. The search space is designed by varying layers in {$2,4,6,8,10$} for each block of ResNet while keeping the sum of layers less than or equal to 18 to maintain architecture size; note that the original ResNet-18 has the four blocks and the parameters are $(4,4,4,4)$. The top 5 searched architectures and their performances are as follows:
>
> Architecture | Mean test acc. $\pm$ std.
> -- | --
> RN18 $(4,4,4,4)$ | 73.27 $\pm$ 0.43
> CRN $(2,2,8,2)$ | 76.10 $\pm$ 1.10
> CRN $(2,2,10,2)$ | 75.65 $\pm$ 1.60
> CRN $(2,10,2,2)$ | **77.01 $\pm$ 0.57**
> CRN $(6,2,6,2)$ | 76.76 $\pm$ 1.58
> CRN $(8,4,2,2)$ | 76.78 $\pm$ 1.72
>
> We have updated our paper by adding this description to Sec. 4.2 and the search results to Sec. C.1.2.
>
> > What's the performance with Architecture RN18(4,4,4,4)?
>
> Thank you for the question. We tested RN18 on all methods in Table 4 after reading your comments. The followings are the results.
>
> Method | RN-18 $(4,4,4,4)$ | Custom RN $(2,10,2,2)$
> -- | -- | --
> Scratch | 73.27 $\pm$ 0.43 | **77.01 $\pm$ 0.57**
> Fine-tuning | 87.75 $\pm$ 0.35 | N/A
> P-SSL | 77.56 $\pm$ 0.62 | **80.86 $\pm$ 0.16**
> PP | 86.09 $\pm$ 0.27 | **88.75 $\pm$ 0.33**
> PP+P-SSL | 87.61 $\pm$ 0.50 | **89.13 $\pm$ 0.23**
>
> This shows that the custom architecture is superior to the original RN18 $(4,4,4,4)$ in all settings, and also indicates that this is a fair comparison among methods. We have added these results in Appendix C.1.2.
>
> > It seems that PP contributes the most in Table 6 but harms the performance in Table 9, any comments on this?
>
> Thank you for the insightful comment and we are sorry for the confusion. We are afraid to note that the baseline of Table 9 is `Fine-tuning` instead of `Scratch` because the `Scratch` setting of the object detection task is hard to train and too slow to converge. That is, Table 9 is a counterfactual experiment that evaluates how practical the proposed method can be compared to the case where the architectures are consistent (i.e., Fine-tuning). In this perspective, PP consistently achieved a competitive performance with Fine-tuning as well as the result is shown in Table 7. Therefore, the conclusions regarding PP are the same for the classification and object detection tasks. We have added this additional description to Sec. 4.7.
>
> > Are the authors going to release the code when the paper get accepted?
>
> Yes. We have already provided the code for reviewers privately (please see the above comment). We plan to upload the code on GitHub when the paper gets accepted.

---

> ### Author Response · Authors · 2022-11-17
> **Gently reminder for phase 1 discussion to Reviewer MbsB (ends Nov. 18)**
>
> Thank you again for your precise and insightful comments on our paper! We sincerely remind you that phase 1 of the discussion period ends on Nov. 18 (Next Friday). Your comments have made our paper's description much clearer. Have your concerns about our proposed method and experiments been addressed? If there are other additional concerns that are significant enough to require revision, it would be greatly appreciated if you could let us know.

---

### Official Review · Reviewer_JCMy · 2022-10-28

**Confidence:** 4
**Correctness:** 4
**Technical Novelty And Significance:** 3
**Empirical Novelty And Significance:** 4
**Recommendation:** 8

**Clarity, Quality, Novelty And Reproducibility:**

The paper is generally clear and is not difficult to follow. The motivations are often re-stated multiple times which makes it a bit confusing. I would request the authors to make this aspect clear-er.

The work quality is quite good. The paper supports its claims with empirical studies mentioned in the appendices.

There is enough details in the paper to reproduce the results.

**Typo**

Fig. 1 caption, (a) …’sampled source label ys’.



**Strength And Weaknesses:**

**Strength**

1. PP combined with P-SSL gives a strong experimental evidence that one can use standard Generative models and their corresponding classifiers in the context of transfer learning.

2. The proposed method seems stable under different target architectures and datasets, as evidenced by various empirical experiments.

3. The proposed method is demonstrated for classification as well as object detection tasks, which makes it more generic.

**Weakness**

The generated samples are then used to learn an encoder representation which is then used to train a P-SSL. Why not use the encoder of generative models to train P-SSL directly, instead of training it on synthetic samples?

**Summary Of The Paper:**

Transfer learning presents different types of problems like non-availability of source data (privacy, costs, etc), or the difference in source and target labels or the target task is different. In this light, the paper proposes to use conditional generative models (CGM) in a two stage proposed method which involves pseudo pretraining (PP) on target architecture on synthesized datasets (CGM),  and then using pseudo semi-supervised learning (P-SSL) on the target data and pseudo samples to tackle the task.

The results show that the approach conditioned on the CGM does provide considerable and stable improvements, specifically as compared to scratch training or knowledge distillation.

Additionally, the empirical experiments reveal that source and target datasets do not necessarily need to be close, and still achieve a good performance gain.

**Summary Of The Review:**

The paper demonstrates, through various experiments, the validity of their claims satisfactorily. For me, the work is quite good and gives a lot of take-aways for further research.

---

> ### Author Response · Authors · 2022-11-11
> **First response to Reviewer JCMy**
>
> Thank you for carefully reading our paper and providing constructive review comments.
>
> ### Replies to weakness
>
> > The generated samples are then used to learn an encoder representation which is then used to train a P-SSL. Why not use the encoder of generative models to train P-SSL directly, instead of training it on synthetic samples?
>
> Thank you for suggesting a very interesting idea. We recognize that your question is asking the reason why we did not directly use the encoders of the source generative models as the initial parameters of the encoders for the target task. Indeed, direct use of the encoder parameters of generative models can help to skip the generating process of synthetic samples in both PP and P-SSL. However, generative models do not necessarily have encoders e.g., we mainly used GANs in the experiments and standard GANs have a discriminator but not an encoder. Furthermore, using specific encoder architectures requires another architecture consistency between the source generative task and target task. Since recent generative models have improved their performance by increasing their network size [1,2], they might not match the requirements of downstream tasks in particular target tasks on restricted embedded devices as we exampled in Section 1. In this perspective, PP and P-SSL, which use synthetic samples, have the advantage of improving target performance regardless of the source architecture.
>
> Nevertheless, the direction using the encoders was curious and thus we experimented with it by assuming BigGAN's discriminator as a pre-trained encoder of target classifiers. We fine-tuned the discriminator by replacing its binary output layer to be a classifier on StanfordCars:
>
> Method (architecture) | Mean test acc. $\pm$ std
> -- | --
> Scratch (WRN-50-2) | 76.92 $\pm$ 2.16
> Finetuning (BigGAN Discriminator) | 16.09 $\pm$ 0.10
> Ours (WRN-50-2) | **91.76 $\pm$ 0.41**
>
> The BigGAN discriminator easily overfitted the target dataset and catastrophically degraded the test performance. It seems difficult to achieve high accuracy by naïvely applying the discriminator as a simple pre-trained encoder. This suggests that the representation of the generative model is quite different from one of the classification tasks and that our method is well suited to absorb such a difference between tasks for knowledge transfer. We have added this discussion and result in Appendix C.1.1.
>
> ### Replies to typo
>
> > Typo; Fig. 1 caption, (a) …’sampled source label ys’.
>
> Thank you for the comment and we are sorry for the typo. We have corrected it in the revision.
>
> If you have any additional questions or concerns, we would be happy to answer them!
>
> ---
> ### Reference
> [1] Brock, Andrew, Jeff Donahue, and Karen Simonyan. "Large scale GAN training for high fidelity natural image synthesis." International Conference on Learning Representation. 2019.
>
> [2] Sauer, Axel, Katja Schwarz, and Andreas Geiger. "Stylegan-xl: Scaling stylegan to large diverse datasets." ACM SIGGRAPH 2022 Conference Proceedings. 2022.

---

> ### Author Response · Authors · 2022-11-17
> **Gently reminder for phase 1 discussion to Reviewer JCMy (ends Nov. 18)**
>
> Thank you again for your constructive and positive comments on our paper! We sincerely remind you that phase 1 of the discussion period ends on Nov. 18 (Next Friday). Does our response correctly address your concerns about using encoders in the generative model? If we have misunderstood any of your comments, we would be very happy to hear from you. Also, we are aware that discussions will continue in phase 2, but if you have any major additional concerns that require revision, please let us know.

---

### Official Review · Reviewer_FAEQ · 2022-11-08

**Confidence:** 4
**Correctness:** 1
**Technical Novelty And Significance:** 1
**Empirical Novelty And Significance:** 1
**Recommendation:** 1

**Clarity, Quality, Novelty And Reproducibility:**

Clarity 3/4
Quality 3/4
Novelty 1/4
Reproducibility 3/4

**Strength And Weaknesses:**

Strength:
This paper is well written, easy to follow and conducts a lot of experiments from scratch, demonstrating the effectiveness of the proposed approach.

Weaknesses:
However, if thinking it carefully, this paper still didn't really solve the problem when condition 2/3 is missing, and condition 1 is trivial.
for condition 1, it is trivial by definition because nowadays most transfer learning tasks assume that source/target class labels are different, that's why pre-training and fine-tuning are so popular.
for condition 2, the authors claimed that source dataset is usually accessible, so they propose a solution that using synthetic data from a generative model G_s with uniform label distribution label_s will be good, however, the exp set up for source dataset is ImageNet/CompCars(in sec 4.6), but this paper is using BigGAN as G_s, which is already powerful enough to represent the missing source dataset. So the question is, what if BigGAN is not available? this paper is just playing a game that basically replacing an unaccessible source dataset with a powerful generative model, So in this regard, this paper never works with condition 2 missing.
then for condition 3, it is obvious that if source dataset (for pretraining) is given, then we can just pretrain it on architecture A_t, and finetune  A_t on target dataset. So in other words, condition 3 is just a byproduct of condition 2, and this paper is just following the same pre-train -> finetune workflow.
In addition, what is the semantic meaning of using an extra unsupervised loss based on [Xie et al., 2020], what will the performance be if without such loss? such ablation study is missing.

It will be more interesting to see that, in this exp set up, if ImageNet is not available as source dataset, can we resort to other less powerful source data set? e.g, using dogs pics to pretrain model and then finetune a task to recognize cars? Not just using a powerful generative model, which could be used to generate a powerful and representative source data set. The problem of how to find a good source data set in this experimental setup is quite trivial, e.g, like privacy concern mentioned in the introduction is interesting but never discussed.

**Summary Of The Paper:**

This paper identifies the key assumptions in transfer learning, addressing an interesting problem that how to solve it when some assumptions does not hold from a systematic perspective. This paper proposes a two-way step, first by pseudo pre-training by using a powerful generative model to generate source data set, then using P-SSL, to fine tune the model combined with unsupervised learning. The extensive empirical exps shows that this approach works "when source dataset is unaccessible".

**Summary Of The Review:**

This paper tries to solve the problem when pretraining dataset is not accessible, but instead using a powerful generative model as a replacement.

---

> ### Author Response · Authors · 2022-11-11
> **First response to Reviewer FAEQ**
>
> Thank you for the review.
>
> ### Overall: Remarks on our problem setting and our method
> > This paper identifies the key assumptions in transfer learning, addressing an interesting problem that how to solve it when some assumptions does not hold from a systematic perspective.
>
> > this paper still didn't really solve the problem when condition 2/3 is missing
>
> > this paper is just playing a game that replacing an unaccessible source dataset with a powerful generative model,
>
> First, let me re-state our problem setting to clarify the conditions.
>
> As stated in Sec. 2 and summarized by Reviewer kCqh, we addressed the transfer learning problem of simultaneously satisfying **all** of the three conditions of (i) no label overlap, (ii) no source dataset access, and (iii) architecture inconsistency, not some conditions. To solve this problem, our main idea is to introduce off-the-shelf pre-trained source generative models $G_s$ to target training, and thus, we assume that $G_s$ is given; we do not access the source dataset. We have confirmed that **our method can achieve higher accuracy than standard fine-tuning** on multiple experiments in Tables 4, 7, 9, and 10. This indicates not only the practicality of our method in our setting but also the advantages of using pre-trained generative models in other settings including fine-tuning. Therefore, our paper is not just replacing a source dataset with a generative model.
>
> ### Replies for comments
>
> > This paper proposes ... pseudo pre-training (PP) by using a powerful generative model to generate source data set
>
> We are afraid to note that PP does not generate source data set statically in advance pre-training (offline manner). PP generates source samples dynamically when training (online manner). Thus. we can avoid massive storage consumptions, unlike offline manner. Further, PP can promote the diversity of generated samples. In fact, Table 17 shows that online PP is superior to offline PP. This can be an advantage of using generative models for transfer learning.
>
> > So the question is, what if BigGAN is not available?
>
> As stated in the above remarks, we assume that $G_s$ is given. Thus, if $G_s$ is not available, then we could not apply our method. On the other hand, we can apply arbitrary conditional generative models for our method including other types of GANs and diffusion probabilistic models. In fact, we have shown the results when using them in Appendix C.3, and confirmed that the diffusion models can be an alternative to BigGANs.
>
> > what is the semantic meaning of using an extra unsupervised loss based on [Xie et al., 2020], what will the performance be if without such loss? such ablation study is missing.
>
> We have discussed this in Sec. 3.2 and C.5.1, and we have provided the ablation study in Sec. C.5.1. The semantic meaning of a pseudo sample $x_{s\leftarrow t}$ is a targe-related sample that does not have exact information of specific single label. Therefore, treating them as unsupervised data enables target models to leverage the source knowledge through soft-pseudo target labels in SSL. In fact, Table 18 in Appendix C.5.1 of the revision shows that the SSL method using a soft label is superior to that using a hard label.
>
> > if ImageNet is not available as source dataset, can we resort to other less powerful source data set? e.g, using dogs pics to pretrain model and then finetune a task to recognize cars?
>
> The experiments in Sec. 4.6 (Table 8) can partially answer your question: $G_s$ on CompCars can help boost the target performance when the target is StanfordCars. For evaluating another case, we tested the case that the source is CompCars and the target is OxfordPets. The results are shown in the following table, which clearly shows that knowledge transfer is difficult when there is less relationship between the target and source. This could be due to Assumption 3.1 is not satisfied well. In fact, the FID is also much larger than the case where ImageNet is the source. Therefore, to maximize the gain from the proposed method, we should choose $G_s$ pre-trained on target-related source datasets.
>
> Method | $\mathcal{D}_s$ :ImageNet | $\mathcal{D}_s$ :CompCars
> -- | -- | --
> PP | 86.31 $\pm$ 0.10 | 74.02 $\pm$ 0.72
> P-SSL | 68.70 $\pm$ 1.21 | 57.44 $\pm$ 4.43
> PP+P-SSL | 89.97 $\pm$ 0.41 | 57.69 $\pm$ 0.85
>
> Metric | $\mathcal{D}_s$ :ImageNet | $\mathcal{D}_s$ :CompCars
> -- | -- | --
> $\text{FID}(\mathcal{D}_{s\leftarrow t}, \mathcal{D}_t)$ | 16.94 | 170.70
>
> > privacy concern mentioned in the introduction is interesting but never discussed
>
> Since the pre-trained source generative models are given in this setting, we do not consider how to train (privacy-preserving or not) the source generative models in this paper. Indeed, the methodology of training for generative models to maintain privacy preservability and the transferability for the downstream tasks is an open problem, so we will explore it in future works.

---

> ### Author Response · Authors · 2022-11-17
> **Gently reminder for phase 1 discussion to Reviewer FAEQ (ends Nov. 18)**
>
> We sincerely remind you that phase 1 of the discussion period ends on Nov. 18 (Next Friday). Have our responses addressed your concerns about the problem setting and the contribution of our paper? We understand that the discussion will continue in phase 2, but if you have other concerns that are significant enough to require a revision of the main paper, we would appreciate it if you would let us know.

---

### Author Response · Authors · 2022-12-09
**Gentle Reminder for Phase 2 Discussion**

Dear all reviewers and ACs,

We sincerely appreciate the reviewers for spending time to read and comment on our paper. We also thank the ACs for carefully conducting the review process.

We would like to remind the reviewers that the discussion period will end on Dec. 12. Your thoughtful comments have greatly enhanced our paper. We have provided detailed responses to the comments and hope that we have addressed your suggestions and questions. If you have any additional comments or feedback on our responses, we would love to hear them. Thank you!

Best regards,

Authors

---

### Decision · Program_Chairs · 2023-01-20

**Decision:**

Reject

**Justification For Why Not Higher Score:**

I don't think the paper is well written enough. It's a simple enough idea, but not well enough explained.

**Justification For Why Not Lower Score:**

N/A

**Metareview: Summary, Strengths And Weaknesses:**

The paper discusses and approach to transfer learning in which the original source dataset is no longer available and target and source label sets are potentially different. The reviewers generally found this an interesting paper, although there does remain significant polarity in the review scores. Primarily this paper introduces a fairly straightforward heuristic, with some empirical justification. In such cases based on fairly simple heuristics I feel the paper should be rock solid in terms of its clarity and reproducibility. Generally there is no consensus that either of these criteria are met.  I personally don't feel that the paper is particularly clear or well written, even after some revision during the rebuttal. Some important topics (such as whether the source and target dimensions should agree) are not addressed and only briefly touched on in the appendix and the notation is overly complex in places. There remain many typos and an overall lack of sufficient care in presentation.